# Improve learning combining crowdsourced labels by weighting Areas Under the Margin

## Abstract

In supervised learning – for instance in image classification – modern massive datasets are commonly labeled by a crowd of workers. The obtained labels in this crowdsourcing setting are then aggregated for training. The aggregation step generally leverages a per worker trust score. Yet, such worker-centric approaches discard each task ambiguity. Some intrinsically ambiguous tasks might even fool expert workers, which could eventually be harmful for the learning step. In a standard supervised learning setting – with one label per task and balanced classes – the Area Under the Margin (AUM) statistic is tailored to identify mislabeled data. We adapt the AUM to identify ambiguous tasks in crowdsourced learning scenarios, introducing the Weighted AUM (WAUM). The WAUM is an average of AUMs weighted by worker and task dependent scores. We show that the WAUM can help discarding ambiguous tasks from the training set, leading to better generalization or calibration performance. We report improvements with respect to feature-blind aggregation strategies both for simulated settings and for the CIFAR-10H crowdsourced dataset.

## 1 Introduction

Crowdsourcing labels for supervised learning has become quite common in the last two decades, notably for image classification with datasets such as CIFAR-10 and Imagenet. Using a crowd of workers is fast, simple (see Figure 1) and less expensive than using experts. Furthermore, aggregating crowdsourced labels instead of working directly with a single one enables modeling the sources of possible ambiguities and directly take them into account in the training pipeline (Aitchison, 2021). With deep neural networks nowadays common in many applications, both the architectures and data quality have a direct impact on the model performance (Müller et al., 2019; Northcutt et al., 2021b) and on calibration (Guo et al., 2017). Yet, depending on the crowd and platform's control mechanisms, the obtained label quality might vary and harm generalization (Snow et al., 2008).

Popular label aggregation schemes take into account the uncertainty related to workers' abilities: for example by estimating confusions between classes, or using a latent variable representing each worker trust (Dawid & Skene, 1979; Kim & Ghahramani, 2012; Sinha et al., 2018; Camilleri & Williams, 2019). This leads to scoring workers without taking into account the inherent difficulty of a task at stake. Inspired by the Item Response Theory (IRT) (Birnbaum, 1968), Whitehill et al. (2009) combined both the task difficulty and the worker's ability in a feature-blind fashion for label aggregation. They only require labels but not the associated features[1]. In the classical supervised learning setting, the labels are said to be *hard* – *i.e.,* a Dirac mass on one class. Multiple crowdsourced labels induce *soft* labels – *i.e.,* probability distributions over the classes – for each task. Our motivation is to identify ambiguous tasks from their associated features, hence discarding hurtful tasks (such as the one illustrated on Figure 2b). Recent work on data-cleaning in supervised learning (Han et al., 2019; Pleiss et al., 2020; Northcutt et al., 2021a) has shown that some images might be too corrupted or too ambiguous to be labeled by humans. Hence, one should not consider these tasks for label aggregation and learning since they might be harmful for generalization.

In this work, we combine task difficulty scores with worker abilities scores, but we measure the task difficulty by incorporating feature information. We thus introduce the Weighted Area Under

---

[1] We use the term task interchangeably with the term feature in this work

Figure 1: Crowdsourcing labels scheme, from label collection using a crowd to training a neural network on aggregated training labels. High ambiguity from either crowd workers or tasks intrinsic difficulty can lead to mislabeled data and harm generalization performance. To illustrate our notation, here the set of task annotated by worker $w_3$ is $\mathcal{T}(w_3) = \{1, 3\}$ while the set of workers annotating the task $x_3$ is $\mathcal{A}(x_3) = \{1, 3, 4\}$.

the Margin (WAUM), a generalization to the crowdsourcing setting of the Area Under the Margin (AUM) (Pleiss et al., 2020). The AUM is a confidence indicator in an assigned label defined for each training task. It is computed as an average of margins over scores obtained along the learning steps, and reflects how a learning procedure struggles to classify a task to an assigned label (see Figures 3 and 5 to visualize how the AUM is connected to the classical margin from the kernel literature). The AUM is well suited when training a neural network (where the steps are training epochs) or other iterative methods. For instance, it has led to better network calibration (Park & Caragea, 2022) using MixUp strategy (Zhang et al., 2018), *i.e.,* mixing tasks identified as simple and difficult by the AUM. The WAUM identifies harmful data points in crowdsourced datasets, so one can prune ambiguous tasks that degrade the generalization. It is a weighted average of workers AUM, where the weights reflect trust scores based on tasks difficulty and workers ability.

## 2 RELATED WORK

Inferring a learning consensus from a crowd is a challenging task. In Table 1, we summarize features used by standard strategies to address such a task. In this work we do not consider methods with prior knowledge on workers, since most platforms do not provide this information[2]. Likewise, we do not rely on ground-truth knowledge for any tasks. Hence, trapping-set or control-items based algorithms like ELICE or CLUBS (Khattak, 2017) do not match our framework. Some algorithms rely on self-reported confidence: they directly ask workers their answering confidence and integrate it in the model (Albert et al., 2012; Oyama et al., 2013; Hoang et al., 2021). We discard such cases for several reasons. First, self-reported confidence might not be beneficial without a reject option (Li & Varshney, 2017). Second, workers have a tendency to be under or overconfident, raising questions on how to present self-evaluation and inferring their own scores (Draws et al., 2021).

The most common aggregation step is majority voting (MV), where one selects the label most often answered by workers. MV does not infer any trust score on workers, thus does not leverage workers abilities. MV is also very sensitive to under-performing workers (Gao & Zhou, 2013; Zhou et al., 2015), to biased workers (Kamar et al., 2015), to spammers (Raykar & Yu, 2011), or to a lack of experts for hard tasks (James, 1998; Gao & Zhou, 2013; Germain et al., 2015). Closely related to MV, naive soft labeling goes beyond *hard labels* (also referred to as *one-hot labels*) by computing the frequency of answers per label. In practice, training a neural network with soft labels improves calibration (Guo et al., 2017) with respect to using hard labels. However, both methods are sensitive to spammers (*e.g.,* workers answer *all* tasks randomly) or workers biases (*e.g.,* workers who answer *some* tasks randomly). Hence, the noise induced by workers labeling might not be representative of the actual task difficulty (Jamison & Gurevych, 2015).

Another class of methods leverages latent variables, defining a probabilistic model on worker's responses. The most popular one, proposed by Dawid & Skene (1979) (DS) estimates a single confusion matrix per worker, as a measure of workers' expertise. The vanilla DS model assumes that a worker answers according to a multinomial distribution, yielding a joint estimation procedure of the error-rates and the soft labels through an Expectation-Maximization (EM) algorithm (see Algorithm 2 in Appendix A). Variants on the original DS algorithm include accelerated versions (Sinha et al., 2018), sparse versions (Servajean et al., 2017), clustered versions (Imamura et al., 2018)

---

[2]For instance, by default Amazon Mechanical Turk https://www.mturk.com/ does not provide it.

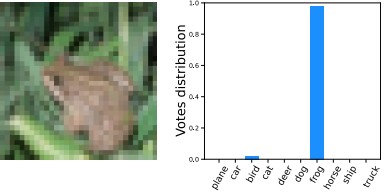 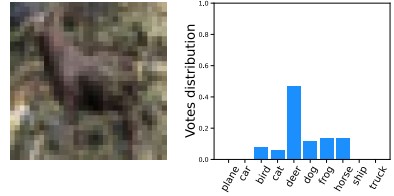

(a) Label `frog` is easy to identify. Only one worker mislabeled this `frog` as a `bird`.

(b) Label `deer` is hard to identify, confused with `horse`, `dog` or other animals available.

Figure 2: Two images from CIFAR-10H dataset (Peterson et al., 2019): the `deer` image is more ambiguous due to the image poor quality. With enough workers, the votes' distribution reflects this ambiguity though in practice there may be only a few workers for some tasks.

among others. Since DS only models workers abilities, Whitehill et al. (2009) have introduced the Generative model of Labels, Abilities, and Difficulties (GLAD) with a task difficulties scores to better handle confusion factors. While DS estimates a matrix of pairwise labels confusion per worker, GLAD estimates (also with EM) a single ability score per worker, and a single difficulty score per task. It is inspired by Item Response Theory (IRT) (Birnbaum, 1968), modeling the workers' probability to answer the true label with a logistic transform of the product of these scores. Following IRT, the difficulty is inferred as a latent variable given the answers, without ever considering the actual task affected to each worker. Other feature-blind aggregation strategies exist using rank-one matrix completion (Ma et al., 2020; Ma & Olshevsky, 2020) or also pairwise co-occurences (Ibrahim et al., 2019). We propose the WAUM to combine the information from a confusion-matrix per worker and a measure of relative difficulty between tasks. It leads to a more accurate judging system and identifies data points harming generalization that should be pruned. Data pruning has been shown to improve generalization by removing mislabeled data (Angelova et al., 2005; Pleiss et al., 2020), possibly dynamically along the learning phase (Raju et al., 2021) or by defining a forgetfulness score (Paul et al., 2021). Finally, Sorscher et al. (2022) have highlighted that data pruning strategies require label information to be successful in the standard supervised setting; we confirm its relevance to the crowdsourcing framework.

Table 1: Summary of vote aggregation algorithms: MV, naive soft, DS, GLAD and WAUM (ours). With the WAUM, we use tasks features to identify and remove harmful tasks.

|  | Soft label | Worker ability | Task difficulty | Use tasks & votes |
|---|---|---|---|---|
| MV | ✗ | ✗ | ✗ | ✗ |
| Naive soft | ✓ | ✗ | ✗ | ✗ |
| DS (vanilla) | ✓ | ✓ | ✗ | ✗ |
| GLAD | ✓ | ✓ | ✓ | ✗ |
| WAUM | ✓ | ✓ | ✓ | ✓ |

# 3 WEIGHTED AREA UNDER THE MARGIN

## 3.1 DEFINITION AND CONSTRUCTION

For any set $\mathcal{S}$, we write $|\mathcal{S}|$ for the cardinality, and for any integer $n$, $[n]$ represents the set $\{1, \ldots, n\}$. We consider classical notation from multi-class classification settings with input from $\mathcal{X}$ (*e.g.,* images) and labels in $[K] = \{1, \ldots, K\}$. Consider a set $\{(x_1, y_1^\star), \ldots, (x_{n_{\text{task}}}, y_{n_{\text{task}}}^\star)\}$ with $n_{\text{task}}$ *i.i.d* tasks and labels. The set of tasks is written as $\mathcal{X}_{\text{train}} = \{x_1, \ldots, x_{n_{\text{task}}}\}$. The underlying distribution is denoted by $\mathbb{P}$, and each pair $(x_i, y_i^\star) \in \mathcal{X} \times [K]$ is a feature/label pair.

In our context, the true labels $y_i^\star$ for $i = 1, \ldots, n_{\text{task}}$ are unobserved (except in simulations); we only collects the proposed labels given by a crowd of $n_{\text{worker}}$ workers (named $w_1, \ldots, w_{n_{\text{worker}}}$). As represented in Figure 1, some workers might annotate more tasks than others, and similarly some tasks might receive more annotations than others. For any $i \in [n_{\text{task}}]$, we write the annotators set given a task $x_i$ as $\mathcal{A}(x_i) = \{j \in [n_{\text{worker}}] : \text{worker } w_j \text{ answered task } x_i\}$. We recover the standard

supervised setting when $|\mathcal{A}(x_i)| = 1$ for all $i \in [n_{\texttt{task}}]$. Similarly, for any $j \in [n_{\texttt{worker}}]$ we write the tasks given a worker $w_j$ as $\mathcal{T}(w_j) = \{i \in [n_{\texttt{task}}] : \text{worker } w_j \text{ answered task } x_i\}$ . For each task $x_i$ and each $j \in \mathcal{A}(x_i)$, we denote $y_i^{(j)} \in [K]$ the label answered by the $j$-th worker, and write $\hat{y}_i$ an estimation of the (possibly soft) label obtained after aggregation. Note that $\hat{y}_i$ belongs to the standard simplex $\Delta_{K-1} = \{p \in \mathbb{R}^K, \sum_{k=1}^{K} p_k = 1, p_k \geq 0, k = [K]\}$ of dimension $K-1$. Our training set has task-wise or worker-wise formulations:

$$\mathcal{D}_{\text{train}} = \bigcup_{i=1}^{n_{\texttt{task}}} \left\{ \left( x_i, \left( y_i^{(j)} \right) \right) \text{ for } j \in \mathcal{A}(x_i) \right\} = \bigcup_{j=1}^{n_{\texttt{worker}}} \left\{ \left( x_i, \left( y_i^{(j)} \right) \right) \text{ for } i \in \mathcal{T}(w_j) \right\} = \bigcup_{j=1}^{n_{\texttt{worker}}} \mathcal{D}_{\text{train}}^{(j)} \; . \tag{1}$$

**DS model.** The Dawid and Skene (DS) model (Dawid & Skene, 1979) aggregates answers and evaluates the workers' confusion matrix to observe where their expertise lies exactly. The confusion matrix of worker $w_j$ is denoted by $\pi^{(j)} \in \mathbb{R}^{K \times K}$ and reflects individual error-rates between pairs of labels. Each individual error-rate $\pi_{\ell k}^{(j)}$, which represents the probability that worker $w_j$ gives label $k$ to a task with true label $\ell$, is estimated using the EM algorithm (see Appendix A, Algorithm 2; the likelihood it maximizes is available in Appendix A, Equation 6). This algorithm is independent of $\mathcal{X}_{\text{train}}$ and only considers the labels. The model assumes that the probability for a task $i$ to have true label $y_i^{\star} = \ell$ follows a multinomial distribution with probabilities $\pi_{\ell \bullet}^{(j)}$ for each worker.

**AUM.** Pleiss et al. (2020) have introduced the AUM in the standard learning setting, where one observes a training set $\mathcal{D}_{\text{train}}$, *i.e.*, with $|\mathcal{A}(x_i)| = 1$ for all $i \in [n_{\texttt{task}}]$. Given a training image and a label $(x, y) \in \mathcal{D}_{\text{train}}$, let $z^{(t)}(x) \in \mathbb{R}^K$ be the associated score vector at epoch $t \leq T$ when learning a neural network on $\mathcal{D}_{\text{train}}$ ($T$ being the number of training epochs). We use the notation $z_{[1]}^{(t)}(x) \geq \cdots \geq z_{[K]}^{(t)}(x)$ for sorting the vector $(z_1^{(t)}(x), \ldots, z_K^{(t)}(x))$ in decreasing order. Denote $\text{softmax}^{(t)}(x) := \text{softmax}(z^{(t)}(x))$ the softmax output of the scores at epoch $t$. Sorting the probabilities in decreasing order such that $\text{softmax}_{[1]}^{(t)}(x) \geq \cdots \geq \text{softmax}_{[K]}^{(t)}(x)$, the AUM reads:

$$\text{AUM}(x, y; \mathcal{D}_{\text{train}}) := \text{AUM}(x, y) = \frac{1}{T} \sum_{t=1}^{T} \left[ \text{softmax}_y^{(t)}(x) - \text{softmax}_{[2]}^{(t)}(x) \right] \in \mathbb{R} \; . \tag{2}$$

Pleiss et al. (2020) uses an average of margins over scores, whereas we instead consider the average over the margin with the softmax output in Equation 2. We have adapted the original AUM by using the $\text{softmax}$ to avoid any scaling issue (as advocated by Ju et al. (2018) in ensemble learning). Another difference is that the margin considered is from Yang & Koyejo (2020), since the corresponding Hinge loss has better theoretical properties, especially for top-$k$ settings[3], than the one used in the original AUM definition (Lapin et al., 2016; Yang & Koyejo, 2020).

During the training phase, the AUM keeps track of the difference between the score assigned to the proposed label and the score assigned to the second largest one. It has been introduced to detect mislabeled observations in a dataset: the higher the AUM, the more confident the prediction is in the assigned label. Hence, the lower the AUM, the more likely the label is wrong. The AUM algorithm is described in Appendix B, Algorithm 4. Finally, note that the AUM computation depends on the chosen neural network and on its initialization: pre-trained architectures could be used, yet any present bias would transfer to the AUM computation.

**WAUM.** The AUM is defined in a standard supervised setting with (hard) labels: we now adapt it to crowdsourced frameworks. Given a $(x_i, y_i^{(j)}) \in \mathcal{D}_{\text{train}} = \mathcal{D}_{\text{train}}^{(1)} \cup \cdots \cup \mathcal{D}_{\text{train}}^{(n_{\texttt{worker}})}$, let $s^{(j)}(x_i) \in [0, 1]$ be a trust factor in the answer of worker $w_j$ for the task $x_i$. The WAUM is then defined as:

$$\text{WAUM}(x_i) = \frac{1}{\sum_{j \in \mathcal{A}(x_i)} s^{(j)}(x_i)} \sum_{j \in \mathcal{A}(x_i)} s^{(j)}(x_i) \text{AUM}(x_i, y_i^{(j)}; \mathcal{D}_{\text{train}}^{(j)}) \in \mathbb{R} \; . \tag{3}$$

The WAUM is the weighted average of the AUMs over each worker's answer with a weighting score $s^{(j)}$ for each task based on workers abilities. The WAUM allows to identify potentially too hard tasks and to remove them from the training set and from the worker's confusion estimation.

---

[3]For top-$k$ accuracy, one could consider $\text{softmax}_{[k+1]}^{(t)}(x)$ instead of $\text{softmax}_{[2]}^{(t)}(x)$ in Equation (2).

The scores $s^{(j)}(x_i)$ consider the impact of the AUM of one answer. It is indeed more informative if an expert worker's AUM indicates a potential uncertainty in the label than a potential error from a poor-quality worker. The architecture used in the AUM computation also needs to be reset between workers to avoid mutual influence. For additional details on AUM and WAUM, see Appendix B. The weights $s^{(j)}$ are obtained *à la* Servajean et al. (2017): each worker has an estimated confusion matrix $\hat{\pi}^{(j)} \in \mathbb{R}^{K \times K}$. Note that the vector $\text{diag}(\hat{\pi}^{(j)}) \in \mathbb{R}^K$ represents the probability for the worker $w_j$ to answer correctly to each task. Moreover, with a neural network classifier, we estimate the probability for the input $x_i \in \mathcal{X}_{\text{train}}$ to belong in each category by $\text{softmax}^{(T)}(x_i)$ – *i.e.,* the probability estimate obtained at the last epoch. As a trust factor, we propose the inner product between the diagonal of the confusion matrix and the softmax vector:

$$s^{(j)}(x_i) = \left\langle \text{diag}(\hat{\pi}^{(j)}), \text{softmax}^{(T)}(x_i) \right\rangle \in [0, 1] \ , \tag{4}$$

that controls the weight of each worker in the $\text{WAUM}(x_i)$ formulation given by Equation (3). This choice of weight is also inspired by the bilinear scoring system of the GLAD strategy as further detailed hereafter. The closer to one, the more we trust the worker for the given task.

**Link with** GLAD. In GLAD (Whitehill et al., 2009), the trust score is modeled as the product $\alpha_j \beta_i$, with $\alpha_j \in \mathbb{R}$ (resp. $\beta_i \in (0, +\infty)$) representing worker ability (resp. task difficulty), *cf.* Appendix A, Algorithm 3. Using DS, we can leverage the full confusion matrices using a scalar product between a quantity related to the worker and another to the task. The diagonal of the confusion matrix $\hat{\pi}^{(j)}$ represents the worker ability and the softmax probabilities represents the task information. This trust score $s^{(j)}(x_i)$ can thus be seen as a multidimensional version of GLAD's trust score.

## 3.2 LABEL AGGREGATION USING THE WAUM.

The WAUM metric aims at identifying training samples that are difficult to classify, and that could be discarded either when aggregating workers' labels or at training. The last step relies on training a neural network thanks to the obtained soft labels, whose architecture is adapted to the task at hand[4].

**Dataset pruning.** Our proposed procedure (Algorithm 1), proceeds as follows. Confusion matrices are estimated for all workers using an estimation step on the full training set $\mathcal{D}_{\text{train}}$. By default we rely on the DS algorithm described in Algorithm 2, but any estimates of workers' confusion matrices can be used instead (see Appendix E). For each worker $w_j$, AUMs are computed for its labeled tasks using Algorithm 4 on $\mathcal{D}_{\text{train}}^{(j)}$, and worker-dependent trust scores for each task $s^{(j)}(x_i)$ with Equation (4). The WAUM in Equation (3) is then computed for each task. Tasks below the quantile of order $\alpha \in [0, 1]$ are then pruned. From the resulting dataset $\mathcal{D}_{\text{pruned}}$ (with tasks $\mathcal{X}_{\text{pruned}}$), we update confusion matrices. We eventually provide soft labels $\hat{y}_i$ for task $x_i \in \mathcal{X}_{\text{pruned}}$ by weighting labels with workers' confidence. A neural network can then be trained on $(x_i, \hat{y}_i)$ for $i \in \mathcal{X}_{\text{pruned}}$. The hyperparameter $\alpha$ (amount of training data points pruned) can be chosen on a validation set, yet choosing $\alpha = 0.1$ or $0.01$ has led to satisfactory results. The output soft labels still contain information regarding human uncertainty, while being less noisy than naive soft labels. They could help improving model calibration (Wen et al., 2021; Zhong et al., 2021), a property often expected for interpretations (Jiang et al., 2012; Kumar et al., 2019).

**Datasets stacking.** In case of time constraints or with few labels per workers, the AUM and worker trust scores can be modified to only train a single network. To do so, one can modify Algorithm 1; Lines 2–5, so the network is trained on $\mathcal{D}_{\text{stack}} := \{(x_i, y_i^{(j)})\}_{i \in [n_{\text{task}}], j \in \mathcal{A}(x_i)}$, the stacking of all workers' training sets. We explore stacking in Appendix E.3 for few labels per worker.

**Refining confusion matrix estimation.** DS might suffer from the curse of dimensionality when the number $K$ of classes is large. Indeed, it relies on estimating $K^2$ coefficients per worker. Possible adaptation depending on the number of workers and how many answers they give are possible. For instance, one can use clustered (workers) confusion matrices in Algorithm 1 as a remedy (Imamura et al., 2018). Another alternative is to leverage the iterative nature of Algorithm 1 and estimate confusion matrices $\hat{\pi}^{(j)}$ only on unambiguous tasks, see Appendix E.2.

---

[4]We describe this choice for each experiment in the following section.

---

**Algorithm 1:** Label aggregation using the WAUM.

---

**Data:** $\mathcal{D}_{\text{train}}$: tasks and crowdsourced labels, $\alpha \in [0, 1]$: cut-off proportion

**Result:** $\{\hat{\pi}^{(j)}\}_{j \in [n_{\text{worker}}]}$: estimated confusion matrices; $(\hat{y}_i)$: tasks' aggregated label

1 **Initialization:** Get confusion matrix estimation $\{\hat{\pi}^{(j)}\}_{j \in [n_{\text{worker}}]}$

2 **for** $j \in [n_{worker}]$ **do**

3      **Train** a neural network on $\mathcal{D}_{\text{train}}^{(j)} = \left\{ \left(x_i, y_i^{(j)}\right) \text{ for } i \in \mathcal{T}(w_j) \right\}$ for $T$ epochs

4      Compute $\text{AUM}(x_i, y_i^{(j)}; \mathcal{D}_{\text{train}}^{(j)})$ using Equation (2) for $i \in \mathcal{T}(w_j)$

5      Compute **trust scores** $s^{(j)}(x_i)$ using Equation (4) for $i \in \mathcal{T}(w_j)$

6 **for** *each task* $x \in \mathcal{X}_{\text{train}}$ **do**

7      Compute $\text{WAUM}(x)$ using Equation (3)

8 Get $q_\alpha$ the WAUM's **quantile threshold** of order $\alpha$ of $(\text{WAUM}(x_i))_{i \in [n_{\text{task}}]}$

9 Define $\mathcal{D}_{\text{pruned}} = \left\{ \left(x_i, \left(y_i^{(j)}\right)_{j \in \mathcal{A}(x_i)}\right) : \text{WAUM}(x_i) \geq q_\alpha \text{ for } i \in [n_{\text{task}}] \right\}$

10 Compute $\{\hat{\pi}^{(j)}\}_{j \in [n_{\text{worker}}]}$ on tasks in $\mathcal{D}_{\text{pruned}}$

11 **Soft labels:** $\hat{y}_i = \frac{\tilde{y}_i}{\sum_{k \in [K]} (\tilde{y}_i)_k} \in \Delta_{K-1}$ with $\tilde{y}_i = \left( \sum_{j \in \mathcal{A}(x_i)} \hat{\pi}_{k,k}^{(j)} \mathbb{1}_{\{y_i^{(j)} = k\}} \right)_{k \in [K]}$ for all $x_i \in \mathcal{X}_{\text{pruned}}$

---

# 4 EXPERIMENTS

In our experiments, we investigate simulations and the CIFAR-10H dataset (Peterson et al., 2019), a crowdsourced dataset with both tasks and workers labels openly accessible. For each aggregation schemes considered, we train a neural network on the soft (except for MV) labels obtained after the aggregation step. We compare our WAUM scheme with MV, Naive Soft, DS and GLAD; see Appendix A for an overview of the methods[5] and Appendix D for details and additional simulations.

**Metrics investigated** After training with the aggregated labels, we report the following performance metrics on test set $\mathcal{D}_{\text{test}}$ (available for simulations and for CIFAR-10H): top-1 accuracy and expected calibration error (ECE) (with $M = 15$ bins as in Guo et al. (2017), see Equation 8 in Appendix C). We also report the train accuracy: $\text{Acc}_{\text{train}}(y^\star, \hat{y}) = \frac{1}{|\mathcal{D}_{\text{train}}|} \sum_{i=1}^{|\mathcal{D}_{\text{train}}|} \mathbb{1}_{\{\arg\max \hat{y}_i = y_i^\star\}}$, *i.e.*, the accuracy of the aggregation method on the training set's true labels (available for simulations and for CIFAR-10H). When dealing with hard labels, the $\arg\max$ can simply be omitted in the definition of $\text{Acc}_{\text{train}}$. For the WAUM, the train accuracy is computed on $\mathcal{D}_{\text{pruned}}$ instead of $\mathcal{D}_{\text{train}}$ as this method does not label tasks detected ambiguous. All results are averaged over 10 repetitions.

**Implementation details** For simulated datasets, the training is performed with a three dense layers' artificial neural network $(30, 20, 20)$ with batch size set to $64$. Moreover, workers are simulated[6] with `scikit-learn` (Pedregosa et al., 2011). For with CIFAR-10H the Resnet-18 (He et al., 2016) architecture is chosen for simplicity (with batch size also set to $64$). To optimize the parameters, we consider an `SGD` optimizer over $150$ training epochs, with initial learning rate of $0.1$, decreasing it by a factor $10$ at epoch $50$ and $100$. Other hyperparameters for `Pytorch`'s (Paszke et al., 2019) `SGD` are `momentum=0.9` and `weight_decay=5e-4`. Our goal is to observe the generalization performance considering feature-blind aggregation strategies against the WAUM. We do not use data augmentation since duplicating mislabeled or poorly labeled tasks can artificially worsen performance (Harutyunyan et al., 2020; Nishi et al., 2021), see Appendix D.1. All experiments are executed on servers equipped with an Nvidia RTX 2080 and Quadro T2000 GPUs.

**Simulated binary dataset: `two_circles`.** We simulate $n_{\text{task}} = 500$ points with `scikit-learn`'s function `two_circles` (with noise $\varepsilon = 0.2$ and scaling factor $0.4$). Our $n_{\text{worker}} = 3$ workers are standard classifiers (see details in Appendix D): a linear Support Vector Machine Classifier (linear SVC), an SVM with RBF kernel (SVC), and a gradient boosted classifier (GBM) with five estimators. Data is split between train ($70\%$) and test ($30\%$) and each simulated worker votes for each task, *i.e.*, for all $x \in \mathcal{X}_{\text{train}}$, $|\mathcal{A}(x)| = n_{\text{worker}}$. Figure 3 illustrates that the AUM identifies tasks misclassified or close to the decision change, for each worker. The WAUM indicates

---

[5]Code is available in the supplementary material; an online repository will be released for the conference.

[6]https://scikit-learn.org/stable/modules/generated/sklearn.datasets.make_circles.html

few potentially ambiguous tasks (low value) and a peak on simple ones (high value), see Figure 3, right part. Tasks with large WAUM are located mostly on the west/south-west of the two circles separation, *i.e.,* where workers disagree the most. Thanks to soft labels and DS confusion matrices estimated on $\mathcal{D}_{\text{pruned}}$, WAUM achieves better test accuracy and expected calibration (in terms of ECE), see Table 2.

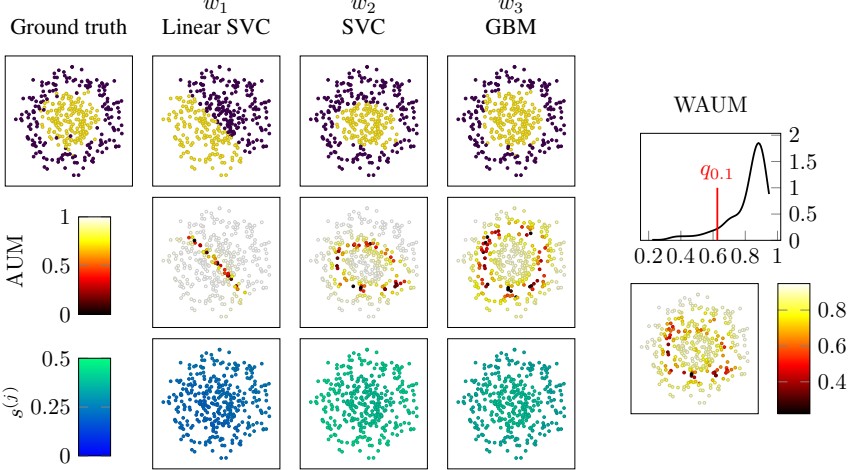

Figure 3: `two_circles`: simulated workers ($w_1, w_2$ and $w_3$) with their associated AUM and normalized trust scores (left) and associated WAUM distributions (right) for $\alpha = 0.1$.

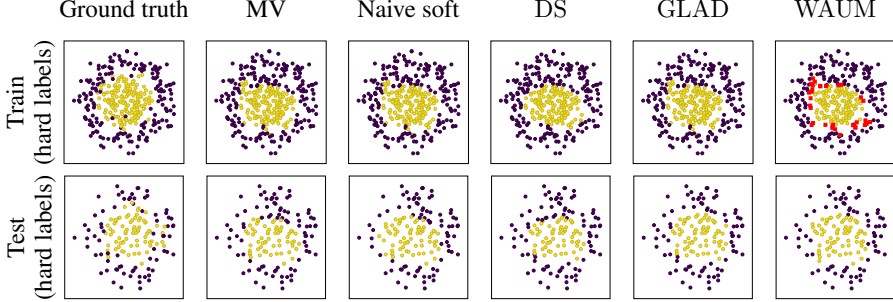

Figure 4: `two_circles`: label predictions on train/test sets provided by a three dense layers' artificial neural network $(30, 20, 20)$ trained on smooth labeled obtained by various aggregation strategies. Points in red are pruned from training by WAUM (here $\alpha = 0.1$). In this binary case, we display binary labels, but soft alternatives can also be visualized (*cf.* Appendix subsection D.2).

Table 2: Aggregation schemes on `two_circles` ($n_{\text{task}} = 500$ tasks, $|\mathcal{A}(x)| = n_{\text{worker}} = 3$. For reference, the best worker is $w_3$ (GBM) with a training accuracy of 0.960 and a test accuracy of 0.880.

| Aggregation | $\text{Acc}_{\text{train}}$ | Test accuracy | ECE |
|---|---|---|---|
| MV | 0.889 | $0.847 \pm 0.004$ | $0.120 \pm 0.006$ |
| Naive soft | 0.889 | $0.810 \pm 0.017$ | $0.091 \pm 0.015$ |
| DS | 0.874 | $0.819 \pm 0.006$ | $0.127 \pm 0.010$ |
| GLAD | 0.889 | $0.847 \pm 0.003$ | $0.123 \pm 0.006$ |
| WAUM($\alpha = 10^{-3}$) | 0.891 | $0.855 \pm 0.001$ | $\mathbf{0.054} \pm 0.006$ |
| WAUM($\alpha = 10^{-2}$) | 0.896 | $\mathbf{0.867} \pm 0.011$ | $\mathbf{0.064} \pm 0.009$ |
| WAUM($\alpha = 10^{-1}$) | 0.930 | $\mathbf{0.863} \pm 0.012$ | $0.075 \pm 0.013$ |
| WAUM($\alpha = 0.25$) | 0.962 | $0.837 \pm 0.016$ | $0.082 \pm 0.012$ |

**Simulated multiclass dataset: `three_circles`.** We adapt the previous example for $K = 3$ classes, simulating 250 tasks per class ($n_{\text{task}} = 750$), and similar workers, each classifying all training tasks. The disagreement area is identified in the north-west area of the dataset as can be seen in Figure 5. Table 3 also shows that pruning too little data ($\alpha$ small) or too much ($\alpha$ large) can mitigate the performance.

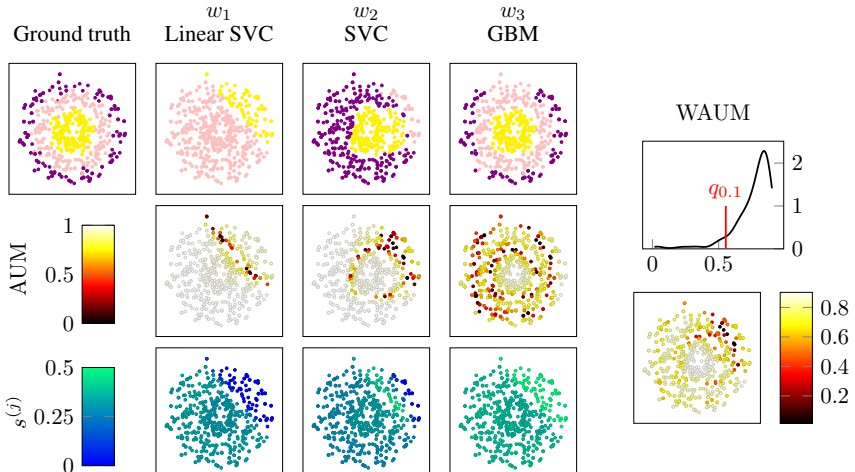

Figure 5: `three_circles`: simulated workers ($w_1$, $w_2$ and $w_3$) with their associated AUM and normalized trust scores (left) and associated WAUM distributions (right) for $\alpha = 0.1$.

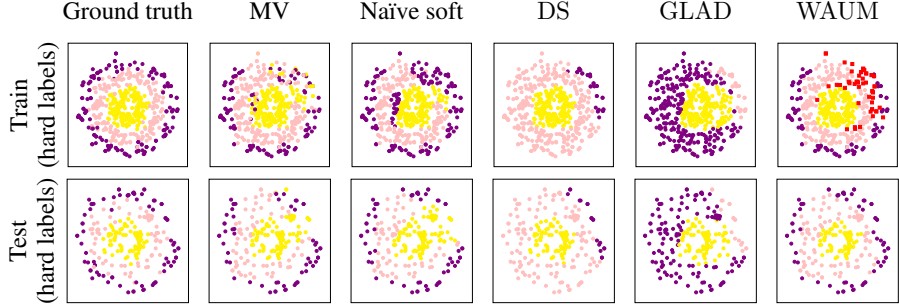

Figure 6: `three_circles`: label predictions on train/test sets provided by a three dense layers' artificial neural network $(30, 20, 20)$ trained on smooth labeled obtained by various aggregation strategies. Points in red are pruned from training by WAUM (here $\alpha = 0.1$).

Table 3: Aggregation schemes on `three_circles` ($n_{\texttt{task}} = 7500$ tasks, $|\mathcal{A}(x)| = n_{\texttt{worker}} = 3$). For reference, the best worker is $w_3$ (GBM) with a training accuracy of 0.920 and a test accuracy of 0.836.

| Aggregation | Acc$_{\text{train}}$ | Test accuracy | ECE |
|---|---|---|---|
| MV | 0.794 | $0.727 \pm 0.026$ | $\mathbf{0.133} \pm 0.027$ |
| Naive soft | 0.794 | $0.697 \pm 0.018$ | $0.178 \pm 0.016$ |
| DS | 0.747 | $0.753 \pm 0.072$ | $0.220 \pm 0.077$ |
| GLAD | 0.549 | $0.578 \pm 0.023$ | $0.356 \pm 0.024$ |
| WAUM($\alpha = 10^{-3}$) | 0.891 | $0.781 \pm 0.042$ | $0.179 \pm 0.014$ |
| WAUM($\alpha = 10^{-2}$) | 0.894 | $0.783 \pm 0.055$ | $0.181 \pm 0.020$ |
| WAUM($\alpha = 10^{-1}$) | 0.936 | $\mathbf{0.806} \pm 0.051$ | $0.186 \pm 0.033$ |
| WAUM($\alpha = 0.25$) | 0.949 | $0.790 \pm 0.042$ | $0.183 \pm 0.025$ |

**CIFAR-10H.** We consider now the CIFAR-10H dataset Peterson et al. (2019). The training part of CIFAR-10H consists of the $10\,000$ tasks of the test set of the original CIFAR-10 dataset (Krizhevsky & Hinton, 2009). In total $n_{\texttt{worker}} = 2571$ workers participated on the Amazon Mechanical Turk platform, each labeling 200 images (20 from each original class), leading to approximately 50 answers per task. We have randomly extracted 500 tasks for a validation set (hence $|\mathcal{D}_{\text{train}}| = 9500$). The final weights output by the network are the one achieving the lowest cross-entropy loss on this set. We test our model on $|\mathcal{D}_{\text{test}}| = 50K$ tasks of the original (hard labeled) train dataset of CIFAR-10. This dataset is notoriously more curated (Aitchison, 2021) than common dataset in the field (but is the only one openly available with both votes and tasks): most difficult tasks were identified and removed at the creation of the CIFAR-10 dataset, resulting in few ambiguities. Table 4 shows that in this simple setting, our data pruning strategy is still relevant, with the choice $\alpha = 0.01$. A visual

influence of the $\alpha$ hyperparameter is available in Appendix D.2 Figure 10. Also notice that in this case, the GLAD algorithm is highly impacted and performs worse than a majority voting.

Furthermore, the WAUM leads to better generalization performance than vanilla DS model or DS model with spammers identification. However, due to the few ambiguous tasks, using naive soft labels can lead to close results for the WAUM, with a slight but consistent gain on our side on the final calibration error. Note that vanilla DS slightly underperformed compared to other aggregation schemes, but using the WAUM we obtain both confusion matrices from DS and aggregated labels with competitive performance.

Table 4: Label recovery, generalization performance and calibration error on the CIFAR-10H dataset by a Resnet-18 (here $\alpha = 0.01$, removing on average 95 tasks).

| Aggregation method | Test accuracy | ECE | $\text{Acc}_{\text{train}}$ |
|---|---|---|---|
| MV | $69.533 \pm 0.84$ | $0.175 \pm 0.00$ | 99.2 |
| Naive soft | $\mathbf{72.149} \pm 2.74$ | $\mathbf{0.132} \pm 0.03$ | 99.2 |
| DS (vanilla) | $70.268 \pm 0.93$ | $0.173 \pm 0.00$ | 99.3 |
| DS (spam identification) | $70.053 \pm 0.81$ | $0.174 \pm 0.0$ | 99.3 |
| GLAD | $66.569 \pm 8.48$ | $0.173 \pm 0.01$ | 99.2 |
| WAUM | $\mathbf{72.747} \pm 1.93$ | $\mathbf{0.124} \pm 0.00$ | 99.2 |

We observe in Table 4 that with labels collected in CIFAR-10H, simple aggregation methods already perform well. Over the 2571 workers, less than 20 are identified as spammers using Raykar & Yu (2011), but we remind that most difficult tasks were removed when creating the CIFAR-10 original dataset. We refer to the CIFAR-10 dataset collection procedure described in the *"labeler instruction sheet"* of Krizhevsky & Hinton (2009, Appendix C) for more information about the written incentives given to workers.

## 5 CONCLUSION AND FUTURE WORK

In this paper, we empirically investigate crowdsourcing aggregation models and how judging systems may impact generalization performance. Most models consider the ambiguity from the workers' perspective (very few consider the difficulty of the task itself) and evaluate workers on hard tasks that might be too ambiguous to be relevant, leading to performance drop. Using a popular model (DS) and small architectures' classifiers, we develop the WAUM, a feature aware metric that improve generalization performance. It also yields a fairer evaluation of workers' abilities and supports recent research on data pruning in supervised datasets. Independently of pruning, the WAUM allows identifying early the images that need extra labelling efforts, or that cannot be correctly labeled at all Extension of the WAUM to more general learning tasks (*e.g.,* top-$k$ classification) would be natural, including sequentially labeling tasks. Indeed, the WAUM could help to identify tasks requiring additional expertise, and guide how to allocate more experts/workers for such identified tasks. Future works need to improve computational efficiency of the WAUM – the vanilla algorithm requires to train one neural network per worker – as initiated with our stacked version (see Appendix E.3). Moreover, adapting the WAUM to imbalanced crowdsourced datasets is of interest to identify potentially too ambiguous images that naturally occurs in open platforms like Pl@ntNet[7]. Last but not least, on the dataset side, providing a challenging dataset (such as the one by Garcin et al. (2021) for instance) to learn in crowdsourcing settings is still missing. Indeed, a dataset with the following properties could greatly foster future research in the field: 1) a varying number of labels per worker, 2) a high number of classes and 3) a subset with ground truth labels to test generalization performance.

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

# A    POPULAR LABEL AGGREGATION TECHNIQUES

Hereafter, we recall

$$\mathcal{D}_{\text{train}} = \bigcup_{i=1}^{n_{\text{task}}} \left\{ \left( x_i, \left( y_i^{(j)} \right) \right) \text{ for } j \in \mathcal{A}(x_i) \right\} \tag{1 revisited}$$

$$= \bigcup_{j=1}^{n_{\text{worker}}} \left\{ \left( x_i, \left( y_i^{(j)} \right) \right) \text{ for } i \in \mathcal{T}(w_j) \right\} = \mathcal{D}_{\text{train}}^{(1)} \cup \cdots \cup \mathcal{D}_{\text{train}}^{(n_{\text{worker}})} \ . \tag{5}$$

In practice, we usually aggregate the crowdsourced labels from $\mathcal{D}_{\text{train}}$ into a probability distribution (soft labels) over the possible classes to train a classifier. For the $i$-th task $x_i$, we write $\hat{y}_i$ for a soft label associated to $x_i$. The loss used to train a (neural network) classifier must be adapted to these soft labels. Here, we have only considered the cross entropy loss:

$$\mathrm{H}(\hat{y}_i, p_i) = - \sum_{k \in [K]} (\hat{y}_i)_k \log(p_i)_k \ ,$$

between the aggregated label $\hat{y}_i$ and the associated predicted probability $p_i$. In our experiments, we have only investigated $p_i = \text{softmax}(x_i)$, *i.e.,* the softmax output associated to the task $x_i$.

In this paper, we compare several aggregation strategies described thereafter: Naive soft, Majority Voting (MV), Dawid and Skene (DS), GLAD, and compared it to our WAUM aggregation.

## A.1    NAIVE SOFT

The naive soft labeling consists on computing the distribution of answered votes:

$$\hat{y}_i = \frac{\tilde{y}_i}{\sum_{k \in [K]} (\tilde{y}_i)_k} \quad \text{with} \quad \tilde{y}_i = \left( \sum_{j \in \mathcal{A}(x_i)} \mathbb{1}_{\{y_i^{(j)} = k\}} \right)_{k \in [K]} \ .$$

## A.2    MAJORITY VOTE

Majority voting simply consists on choosing the most answered label (and can be seen as an $\arg\max$ of the naive soft):

$$\hat{y}_i = \underset{k \in [K]}{\arg\max} \left( \sum_{j \in \mathcal{A}(x_i)} \mathbb{1}_{\{y_i^{(j)} = k\}} \right) \ .$$

## A.3    DAWID AND SKENE

The Dawid and Skene (DS) (Dawid & Skene, 1979) model aggregates answers and evaluates the workers' confusion matrix to observe where their expertise lies exactly. Let us introduce $\rho_\ell$ the prevalence of each label in the dataset, *i.e.,* the probability that a task drawn at random has label $\ell \in [K]$. Following standard notations, we also write $\{T_{i\ell}, \ i \in [n_{\text{task}}]\}$ the indicator variables for task $i$ that is 1 if the task has label $y^\star = \ell$. Finally, let $\pi_{\ell k}^{(j)}$ be the probability for worker $j$ to select label $k$ when $y^\star = \ell$. The model's likelihood is:

$$\prod_{i \in [n_{\text{task}}]} \prod_{\ell \in [K]} \left[ \rho_\ell \prod_{j \in [n_{\text{worker}}]} \prod_{k \in [K]} \left( \pi_{\ell k}^{(j)} \right) \right]^{T_{i\ell}} \ . \tag{6}$$

Using the EM Algorithm 2, we obtain maximum likelihood estimations of $\pi^{(j)}$ and $\rho_\ell$ for $j \in [n_{\text{worker}}]$ and $\ell \in [K]$. The convergence criterion is often chosen as the likelihood's variability between two iterations.

The DS model has been adapted to different settings (sparsity, worker clustering, accelerations) (Servajean et al., 2017; Imamura et al., 2018; Sinha et al., 2018) to overcome limitations mostly due to the estimation of the confusion matrices of size $K^2$ for each worker. We discuss these possible alternatives in Appendix E.

---

**Algorithm 2:** EM algorithm DS model.

---

**Data:** $\mathcal{D}_{\text{train}}$: crowdsourced dataset
**Result:** $\{\hat{\pi}^{(j)}\}_{j\in[n_{\text{worker}}]}$: estimated confusion matrices; $(\hat{y}_i)_{i\in[n_{\text{task}}]}$: tasks soft labels

1 **Initialization:** $\forall i \in [n_{\text{task}}], \forall \ell \in [K], \ \hat{T}_{i\ell} = \frac{1}{|\mathcal{A}(x_i)|}\sum_{j\in\mathcal{A}(x_i)} \mathbb{1}_{\{y_i^{(j)}=\ell\}}$

2 **while** *Convergence not achieved* **do**

    // **M-step:** Get $\hat{\pi}$ and $\hat{\rho}$ assuming $\hat{T}$s are known

3        $\forall(\ell,k)\in[K]^2, \ \hat{\pi}_{\ell k}^{(j)} \leftarrow \dfrac{\sum_{i\in[n_{\text{task}}]}\hat{T}_{i\ell}}{\sum_{k\in[K]}\sum_{i'\in[n_{\text{task}}]}\hat{T}_{i'\ell}}$

4        $\forall\ell\in[K], \ \hat{\rho}_\ell \leftarrow \frac{1}{n_{\text{task}}}\sum_{i\in[n_{\text{task}}]}\hat{T}_{i\ell}$

    // **E-step:** Estimate $\hat{T}$s with current $\hat{\pi}$ and $\hat{\rho}$

5        $\forall i\in[n_{\text{task}}], \forall\ell\in[K], \ \hat{T}_{i\ell} = \dfrac{\prod_{j\in\mathcal{A}(x_i)}\prod_{k\in[K]}\hat{\rho}_\ell\big(\hat{\pi}_{\ell k}^{(j)}\big)}{\sum_{\ell'\in[K]}\prod_{j'\in\mathcal{A}(x_i)}\prod_{k'\in[K]}\hat{\rho}_{\ell'}\big(\hat{\pi}_{\ell'k'}^{(j')}\big)}$

6 **Labels:** $\forall i\in[n_{\text{task}}], \ \hat{y}_i = \hat{T}_{i\bullet}\in\mathbb{R}^K$

---

## A.4 GLAD

We recall the GLAD (Whitehill et al., 2009) algorithm in the binary setting[8]. A modeling assumption is that the $j$-th worker labels correctly the $i$-th task with probability given by

$$\mathbb{P}(y_i^{(j)} = y_i^\star|\alpha_j,\beta_i) = \frac{1}{1+e^{-\alpha_j\beta_i}} \ ,$$

with $\alpha_j\in\mathbb{R}$ the worker's expertise: $\alpha_j < 0$ implies misunderstanding, $\alpha_j = 0$ an impossibility to separate the two classes and $\alpha_j > 0$ a valuable expertise. The coefficient $1/\beta_i\in\mathbb{R}_+$ represents the task's intrinsic difficulty: if $1/\beta_i\to 0$ the task is trivial; on the other side when $1/\beta_i\to +\infty$ the task is very ambiguous. Parameters $(\alpha_j)_{j\in[n_{\text{worker}}]}$ and $(\beta_i)_{i\in[n_{\text{task}}]}$ are estimated using an EM algorithm as described in Algorithm 3.

---

**Algorithm 3:** EM algorithm GLAD model.

---

**Data:** $\mathcal{D}_{\text{train}}$: crowdsourced dataset
**Result:** $\alpha = \{\alpha_j\}_{j\in[n_{\text{worker}}]}$: worker abilities, $\beta = \{\beta_i\}_{i\in[n_{\text{task}}]}$: task difficulties,
        $(\hat{y}_i)_{i\in[n_{\text{task}}]}$: aggregated labels

1 **while** *Convergence not achieved* **do**

2     **E-step**                     // Estimate probability of $y_i^\star$

3        $\forall i\in[n_{\text{task}}], \ \mathbb{P}(y_i^\star|\{y_i^{(j)}\}_i,\alpha,\beta_i)\propto\mathbb{P}(y_i^\star)\prod_j\mathbb{P}(y_i^{(j)}|y_i^\star,\alpha_j,\beta_i)$

4     **M-step**                            // Maximization

5        Maximize auxiliary function $Q(\alpha,\beta)$ in Equation (7) with respect to $\alpha$ and $\beta$

---

The auxiliary function for the binary GLAD model is:

$$Q(\alpha,\beta) = \mathbb{E}[\log\mathbb{P}(\{y_i^{(j)}\}_{ij},\{y_i^\star\}_i)] = \sum_i\mathbb{E}[\log\mathbb{P}(y_i^\star)] + \sum_{ij}\mathbb{E}[\log\mathbb{P}(y_i^{(j)}|y_i^\star,\alpha_j,\beta_i)] \ . \quad (7)$$

Following Whitehill et al. (2009), denote $p^k = \mathbb{P}(y_i^\star = k|\{y_i^{(j)}\}_{ij},\alpha,\beta)$, then one can obtain partial derivatives of the function $Q$ with respect to $\alpha_j$ and $\beta_i$ as follows:

$$\frac{\partial Q}{\partial\alpha_j} = \sum_i(p^1 y_i^{(j)} + p^0(1-y_i^{(j)}) - \mathbb{P}(y_i^{(j)} = y_i^\star|\alpha_j,\beta_i))\beta_i \ ,$$

$$\frac{\partial Q}{\partial\alpha_j} = \sum_j(p^1 y_i^{(j)} + p^0(1-y_i^{(j)}) - \mathbb{P}(y_i^{(j)} = y_i^\star|\alpha_j,\beta_i))\alpha_j \ .$$

Setting the partial derivatives of the function $Q$ to zero, one can solve iteratively the nonlinear system of equations. See Whitehill et al. (2009)[Supplementary material] for additional details.

---

[8]we provide few more details at the end of the paragraph for multi-class settings

An extension to the multiclass setting is given by Whitehill et al. (2009) under the following assumption: the distribution over all incorrect labels is supposed uniform. However, this is not verified in many practical cases, as can be seen for example in Figure 2 where the `deer` label is only mistaken with labels referring to other animals and not with vehicles. We have used the implementation from https://github.com/notani/python-glad to evaluate the GLAD performance in our experiments. Instead of setting the partial derivatives to zero and solving the nonlinear system, it maximizes the function $Q$ with respect to $\alpha$ and $\beta$ using the conjugate gradient algorithm.

## B  ADDITIONAL DETAILS ON AUM AND WAUM

AUM **computation in practice.**    We recall in Algorithm 4 how to compute the AUM in practice for a given training set $\mathcal{D}_{\text{train}}$. This step is used within the WAUM (label aggregation step). Overall, with respect to training a model, computing the AUM, requires an additional cost: $T$ training epochs are needed to record margins for each task. This usually represents less than twice the original time budget. We recall that $\text{softmax}^{(t)}(x_i)$ is the softmax output of the predicted scores for the task $x_i$ at iteration $t$.

---

**Algorithm 4:** AUM algorithm.

**Data:** $\mathcal{D}_{\text{train}} = (x_i, y_i)_{i \in [n_{\text{task}}]}$: training set with $n_{\text{task}}$ task/label couples, $T$: total epochs.
**Result:** $(\text{AUM}(x_i, y_i))_{i \in [n_{\text{task}}]}$: tasks' AUM.

1  **for** $t = 1, \ldots, T$ **do**
2      **Train** the neural network for epoch $t$ using $\mathcal{D}_{\text{train}}$
3      **for** $i \in [n_{task}]$ **do**
4          **Record softmax** output $\text{softmax}^{(t)}(x_i) \in \Delta_{K-1}$
5          **Compute margin** $M^{(t)}(x_i, y_i) = \text{softmax}_{y_i}^{(t)}(x_i) - \text{softmax}_{[2]}^{(t)}(x_i)$
6  $\forall i \in [n_{\text{task}}],\ \text{AUM}(x_i, y_i; \mathcal{D}_{\text{train}}) = \frac{1}{T} \sum_{t \in [T]} M^{(t)}(x_i, y_i)$ .

---

**Example 1** (Confusion matrix update). *The* WAUM *procedure estimates each worker's confusion matrix using a DS-like model first on $\mathcal{D}_{train}$, and then on $\mathcal{D}_{pruned}$. A naturally arising question is then: does pruning these few tasks selected with lowest* WAUM *really has an impact on the estimates matrices $\hat{\pi}^{(j)}$? Let us go back to the* `three_circles` *simulated dataset from Section 4 and compare the confusion matrices on $\mathcal{D}_{train}$ denoted $\pi_{train}^{(j)}$ and on $\mathcal{D}_{pruned}$ denotes $\pi_{pruned}^{(j)}$ for our three simulated workers. We also represent in Figure 7 the variation between the estimated confusion matrices on $\mathcal{D}_{train}$ and $\mathcal{D}_{pruned}$. Figure 7 shows that on the* `three_circles` *dataset, removing ambiguous training data with a cut-off parameter $\alpha = 0.1$ lead to significant changes in the estimated error rates in the confusion matrices. It decreased the confidence in $w_1$ to find the innermost class, and increased the confidence in the best worker $w_3$.*

**Example 2** (Trust factors). *Assume $K = 3$, $n_{worker} = 2$ and confusion matrices:*

$$\hat{\pi}^{(1)} = \begin{pmatrix} 1 & 0 & 0 \\ 0.5 & 0.5 & 0 \\ 0.5 & 0.5 & 0 \end{pmatrix} \quad and \quad \hat{\pi}^{(2)} = \begin{pmatrix} 0 & 0.5 & 0.5 \\ 0.5 & 0.5 & 0 \\ 0 & 0 & 1 \end{pmatrix} \ .$$

*Given a task $x$ answered by both workers and an arbitrary neural network trained with $T$ epochs such that $\text{softmax}^{(T)}(x) = (0.5, 0.5, 0)$, then $s^{(1)}(x) = 0.75$ and $s^{(2)}(x) = 0.25$. Thus, the averaged margin with the label proposed by worker $w_1$ counts three times more than the one from worker $w_2$ as the network ambiguity only comes from the first two labels where worker $w_1$ is more competent.*

**Example 3** (Extreme cases for WAUM aggregations). *Aggregation strategies often return soft labels, but in a context with low uncertainty, hard label (say Dirac distribution) might be popular too. With high uncertainty, the estimated label distribution tend to be uniform over the $K$ classes. Such phenomenon naturally arises with* WAUM *aggregation. In the following we illustrate when these extreme cases occur in the binary setting.*

*Using Algorithm 1, the aggregated label $\hat{y}$ of a task in $\mathcal{D}_{pruned}$ is a Dirac distribution (perfect consensus) if all workers, independently of their confusions, answer the same label. On the other extreme,*

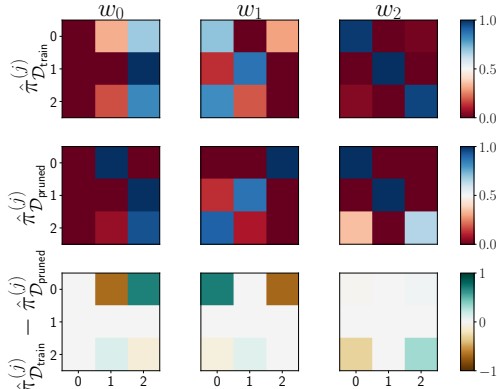

Figure 7: Confusion matrices of the three simulated workers on the `three_circles` dataset before and after pruning tasks selected with lowest WAUM with $\alpha = 0.1$.

*soft labels representing uniform distributions can be obtained by many configurations. In a binary setting ($K = 2$), picking $n_{worker} = 3$ such that $y^{(1)} = y^{(2)} = 1$, $y^{(3)} = 2$ and*

$$\hat{\pi}^{(1)} = \begin{pmatrix} 0.25 & 0.75 \\ 0.4 & 0.6 \end{pmatrix}, \qquad \hat{\pi}^{(2)} = \begin{pmatrix} 0.25 & 0.75 \\ 0 & 1 \end{pmatrix} \quad and \quad \hat{\pi}^{(3)} = \begin{pmatrix} 0.5 & 0.5 \\ 0 & 1 \end{pmatrix} ,$$

*results in uniform soft labels: $\hat{y} = (\frac{1}{2}, \frac{1}{2}) \in \Delta_1$. Similar compensations occur for $K > 2$.*

### B.1 AUM and WAUM variations

In this part we show the impact of the choice of the criterion for identifying the difficulty of a task, or the likelihood that a label is wrong. For that we compare on CIFAR-10H, four variants of AUM for identifying wrong or ambiguous labels. Each line represents one of the 10 classes in CIFAR-10H, where the images are the 10 worst images by class for each AUM variant.

- our proposed AUM as described in Equation (2) following Yang & Koyejo (2020)
- the original AUM by as described in Pleiss et al. (2020, Eq. (1))
- the WAUM using our proposed formulation of the AUM and following Yang & Koyejo (2020)
- the WAUM using the original AUM by as described in Pleiss et al. (2020, Eq. (1))

The images surrounded by a red box in Figure 8 are possible wrong or undecidable labels according to our personal judgment. The conclusion are as follows. First, note that the AUM variants without using the crowdsourced labels are pretty similar for our proposed and the original (only one image detected as wrong or ambiguous, for the original, *cf.* Figure 8c, and none for Figure 8a). For the WAUM variants both criterion provided many more discoveries of wrong or ambiguous labels (seven and eight respectively for the WAUM using our proposed formulation of the AUM, see Figure 8c and following Yang & Koyejo (2020) and for the one using the original formulation, see Figure 8d).

## C Reminder on calibration of neural networks

Hereafter, we propose a reminder on neural networks calibration metric defined in Guo et al. (2017). Calibration measures the discrepancy between the accuracy and the confidence of a network. In this context, we say that a neural network is perfectly calibrated if it is as accurate as it is confident. For each task $x \in \mathcal{X}_{\text{train}} = \{x_1, \dots, x_{n_{\text{task}}}\}$, let us recall that an associated predicted probability distribution is provided by $\text{softmax}(x) \in \Delta_{K-1}$. Let us split the prediction interval $[0, 1]$ into $M$ bins $I_1, \dots, I_M$ of size $1/M$: $I_m = (\frac{m-1}{M}, \frac{m}{M}]$. Following Guo et al. (2017), we denote $B_m =$

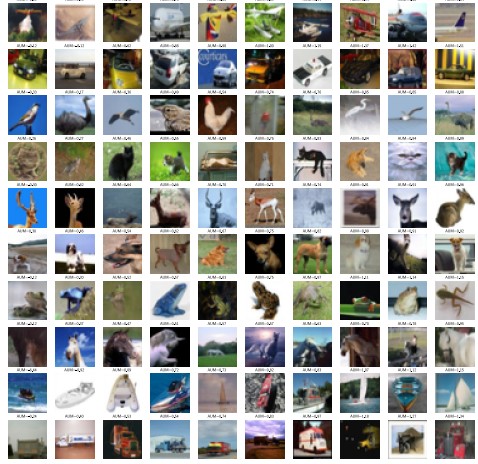

(a) Our proposed AUM as described in Equation (2) following Yang & Koyejo (2020)

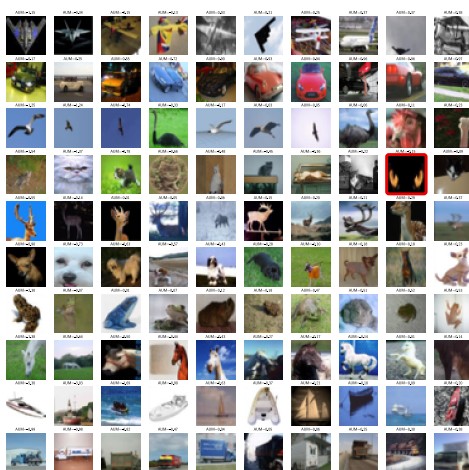

(b) Original AUM, as described in Pleiss et al. (2020)

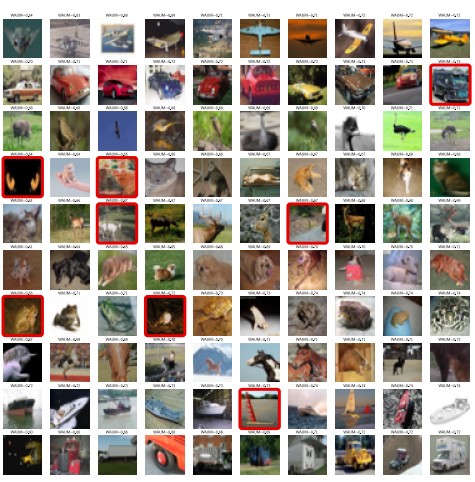

(c) The WAUM using our proposed formulation of the AUM and following Yang & Koyejo (2020)

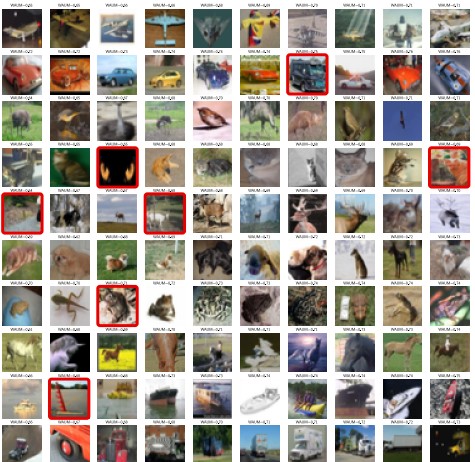

(d) The WAUM using the original AUM by as described by Pleiss et al. (2020)

Figure 8: CIFAR-10H: Worst images detected by class, for four variants of AUM. Each line represents for each of the 10 classes, the 10 worst AUM or WAUM: a) our proposed AUM as described in Equation (2) following Yang & Koyejo (2020) b) original AUM by Pleiss et al. (2020), c) WAUM using our proposed formulation of the AUM and following Yang & Koyejo (2020) and d) the WAUM using the original AUM by as described in Pleiss et al. (2020, Eq. (1)). The images surrounded by a red box are possible wrong or undecidable labels

$\{x \in \mathcal{X}_{\text{train}} : \text{softmax}_{[1]}(x) \in I_m\}$ the task whose predicted probability is in the $m$-th bin[9]. We recall that the accuracy of the network for the samples in $B_m$ is given by:

$$\text{acc}(B_m) = \frac{1}{|B_m|} \sum_{i \in B_m} \mathbb{1}_{\{\text{softmax}_{[1]}(x_i) = y_i\}} \ .$$

We can now define the empirical confidence on the same $B_m$ by:

$$\text{conf}(B_m) = \frac{1}{|B_m|} \sum_{i \in B_m} \text{softmax}_{[1]}(x_i) \ .$$

---

[9]Remember that with our notation $\text{softmax}_{[1]}(x) = \arg\max_{k \in [K]} (\text{softmax}(x))_k$, with ties broken at random.

Finally, the expected calibration error (ECE) reads:

$$\text{ECE} = \sum_{m=1}^{M} \frac{|B_m|}{n_{\texttt{task}}} |\text{acc}(B_m) - \text{conf}(B_m)| \quad . \tag{8}$$

A neural network is said *perfectly calibrated* if $\text{ECE} = 0$, thus if the accuracy equals the confidence for each subset $B_m$.

## D  LEARNING ON SIMULATED CROWDSOURCED DATASETS

In this section, we provide some details about the experiments on simulated datasets `two_circles` and `three_circles` described in Section 4. We also consider two extra experiments showing some limitations of the pruning step.

We consider simulated datasets and workers represented by popular classifiers: a linear SVC, an SVC with RBF kernel and a gradient boosted classifier (with five estimators). To induce more ambiguity (and avoid too similar workers), one-shot learners are classifiers with maximum iteration set to 1 in the learning phase. Other hyperparameters are set to `scikit-learn`'s default values[10].

The `two_circles` dataset is simulated with the `make_circles` from `scikit-learn`. The noise $\varepsilon = 0.2$ and the factor is set to 0.4. For the three_circles's dataset, we use the same `scikit-learn` function twice, with two different factors: 0.3 and 0.7 creating each circle ring.

### D.1  LABEL NOISE AND DATA AUGMENTATION

The classification dataset `two_circles` is also generated using `make_circles`, with $\varepsilon = 0.2$ indicating the standard deviation of the noise added to each task. We also use a factor of 0.4 to scale between the two circles. We generate $n_{\texttt{task}} = 500$ tasks addressed by our workers. Workers are simulated as in Section 4 with answers stated in Figure 3. Here, we show with the simulation setting of Section 4 that data augmentation with noisy tasks could hurt the test accuracy.

Each task $x \in \mathcal{X}_{\text{train}}$ is labeled by all workers, and augmented by adding a Gaussian noise with probability $p \in \{0.1, 0.25, 0.5, 0.75\}$ as follows (on each batch):

$$x_{\text{aug}} = x + \mathcal{N}(0, 0.5^2) \quad .$$

We then train a three dense layers' artificial neural network $(30, 20, 20)$ neural network on aggregated soft labels from naive soft and WAUM methods and compare the test accuracy in Table 5. Duplicating too much noisy labels at learning can thus harm the generalization performance.

Table 5: Test accuracy after learning with augmented data for the `two_circles` dataset with $n_{\texttt{task}} = 500$ tasks and $|\mathcal{A}(x)| = n_{\texttt{worker}}$ by a three dense layers' artificial neural network $(30, 20, 20)$. Each $x \in \mathcal{X}_{\text{train}}$ is augmented with probability $p$. Results are averaged over 10 repetitions.

| $p$ | 0 | 0.1 | 0.25 | 0.5 | 0.75 |
|---|---|---|---|---|---|
| Naive soft | $\mathbf{0.810} \pm 0.02$ | $0.619 \pm 0.01$ | $0.592 \pm 0.02$ | $0.536 \pm 0.00$ | $0.482 \pm 0.02$ |
| WAUM($\alpha = 0.01$) | $\mathbf{0.863} \pm 0.01$ | $0.795 \pm 0.01$ | $0.782 \pm 0.01$ | $0.736 \pm 0.01$ | $0.684 \pm 0.01$ |

### D.2  OTHER SIMULATED CROWDSOURCED DATASETS

We provided here more datasets to illustrate some aspects of our proposed WAUM, as well as its limits. In all cases, the data is split into train (70%) and test (30%) sets.

- The `two_circles` dataset, as introduced in Section 4, to highlight the ambiguity visualization.

---

[10]For instance, the squared-hinge loss function is penalized with $\ell^2$ regularization with strength set to 1 for the linear SVC and the SVC), the gradient boosted classifier uses the multinomial deviance as loss, and the maximum depth (number of nodes in trees) defaults to 3.

- The `three_circles` dataset, introduced in Section 4, to show the visual impact of the parameter $\alpha$ on the WAUM.

- The `make_classification` dataset, that consists on two clusters. It provides a case where pruning is not relevant and all methods achieve similar generalization.

- The `two_moons` dataset with an intrinsic difficulty that is highly relevant for the final decision, in this setting, pruning is unadvised as it can hurt generalization performance.

The first dataset does not present any significant difficulty thus leading to similar performance across aggregated labels. The goal with the `two_moons` dataset is to showcase a limitation of the WAUM: removing data should not be done at the cost of removing inherent structure. We recall that the three simulated workers are a Linear Support Vector Classifier (Linear SVC), a one-shot radial basis function SVC (SVC) and a gradient boosting classifier using trees with five estimators (GBM).

**The `two_circles` dataset: ambiguity visualization.** The workers' answers here are unchanged from Figure 3. This part shows the ambiguity in the soft labels estimated with $\mathcal{D}_{\text{train}}$ using the different aggregation strategies. We also represent the ambiguity in the output distribution. Each point represents a task $x_i$, and the associated label $\hat{y}_i$ is represented with the color. Indeed, $\hat{y}_i \in \Delta_1$, thus knowing the probability to belong in one class, the other probability can easily be inferred.

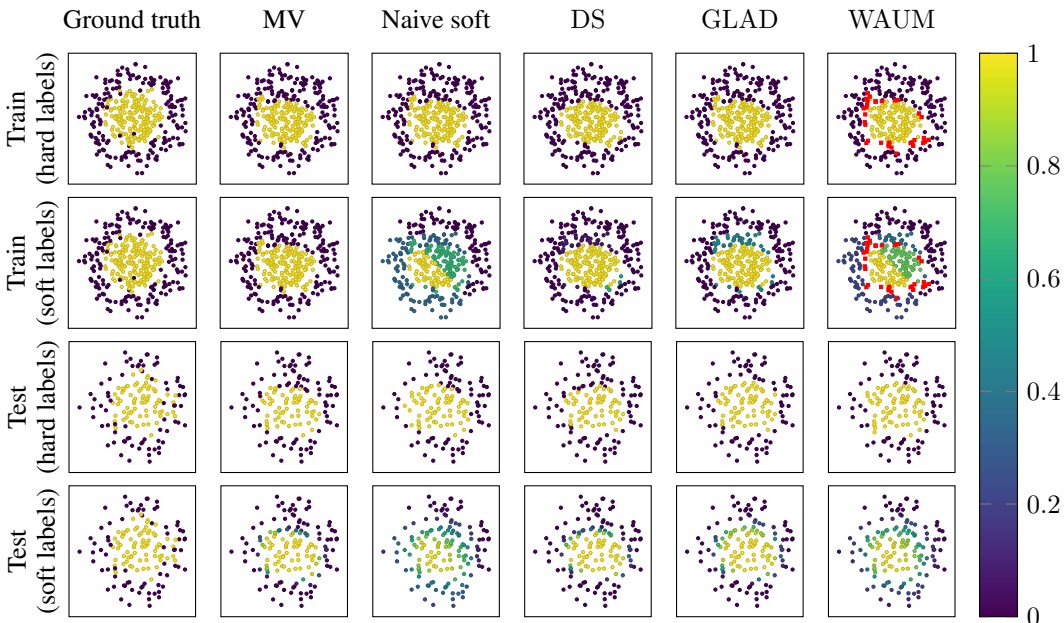

Figure 9: `two_circles`: soft labels (probabilities) predictions on train/test sets provided by a three dense layers' artificial neural network $(30, 20, 20)$ trained on smooth labeled obtained by various aggregation strategies. Points in red are pruned from training by WAUM (here $\alpha = 0.1$). Each point represents a task $x_i$, and its color is the probability to belong in class 1. This lets us visualize the ambiguity in the soft training aggregated labels, but also in the resulting predictions by the neural network.

Figure 9 shows that most training labels with DS and GLAD do not induce much ambiguity. This lack of ambiguity is then reflected in the test output with more confident probabilities. However, the WAUM labels have less uncertainty than the Naive soft ones, especially in the south-west region. There, the test probabilities also reflect this confidence, while keeping the uncertainty near the decision boundary similar to the one by the Naive soft strategy. Overall, the WAUM aggregation removed much of the unneeded uncertainty and kept the uncertainty on very ambiguous tasks.

**The `three circles` dataset: influence of $\alpha$.** In Table 3, we show on the `three circles` simulated dataset that we get better performance with the WAUM using hyperparameter $\alpha = 0.1$. We visually compare the influence of this quantile hyperparameter on the pruning, in Figure 10.

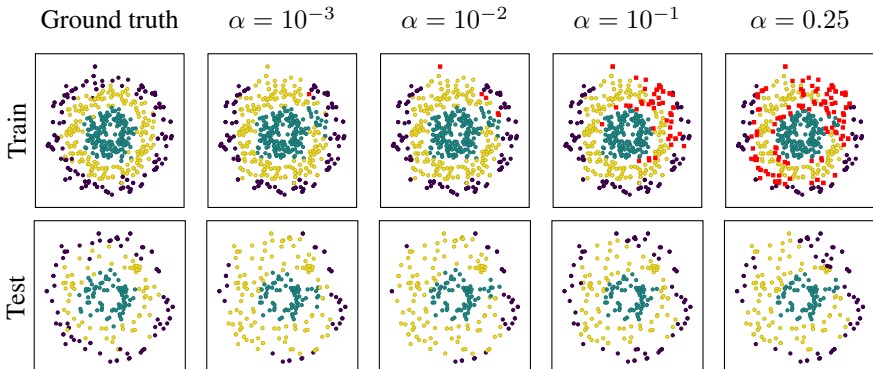

Figure 10: Influence of $\alpha$ on the pruning step. Red dots indicate data points pruned from the training set, at level $q_\alpha$ in the WAUM (see line 8 in Algorithm 1). We consider ($\alpha \in \{10^{-3}, 10^{-2}, 10^{-1}, 0.25\}$). The closer $\alpha$ is to 1, the more training tasks are pruned from the training set (and the worse the performance). The neural network used for predictions is a three dense layers' $(30, 20, 20)$, as for other simulated experiments.

**The `make_classification` dataset: a case where pruning is useless.** We simulate $n_{\texttt{task}} = 500$ tasks using `make_classification` from `scikit-learn` using two clusters per class (here $K = 2$) and split the data in train/test with a test size of 0.3. We consider a class separation factor of 1.5 on the hypercube. With this dataset, all methods achieve similar performance.

Table 6: Training and test accuracy depending on the aggregation method used for the `make_classification`'s dataset with $n_{\texttt{task}} = 500$ points used for training a three dense layers' artificial neural network $(30, 20, 20)$. For reference the best workers are $w_1$ and $w_3$ with respective training accuracies of 0.786 and 0.790 and test accuracies of 0.770 and 0.660.

| Aggregation | $\text{Acc}_{\text{train}}$ | Test accuracy | ECE |
|---|---|---|---|
| MV | 0.923 | $0.907 \pm 0.000$ | $0.085 \pm 0.000$ |
| Naive soft | 0.923 | $0.906 \pm 0.002$ | $0.160 \pm 0.012$ |
| DS | 0.920 | $0.886 \pm 0.000$ | $0.108 \pm 0.002$ |
| GLAD | 0.926 | $0.893 \pm 0.004$ | $\mathbf{0.076} \pm 0.004$ |
| WAUM($\alpha = 10^{-3}$) | 0.928 | $0.897 \pm 0.006$ | $\mathbf{0.078} \pm 0.013$ |
| WAUM($\alpha = 10^{-2}$) | 0.933 | $0.901 \pm 0.002$ | $\mathbf{0.078} \pm 0.012$ |
| WAUM($\alpha = 10^{-1}$) | 0.965 | $0.889 \pm 0.007$ | $0.084 \pm 0.016$ |
| WAUM($\alpha = 0.25$) | 0.977 | $0.899 \pm 0.011$ | $0.084 \pm 0.015$ |

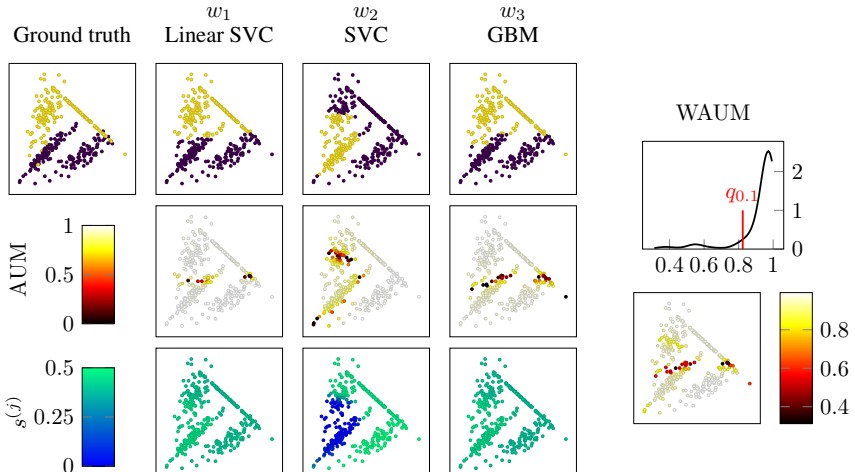

Figure 11: Simulated workers with associated AUM and normalized trust scores on the `make_classification` dataset. The hyperparameter $\alpha$ is set to 0.1.

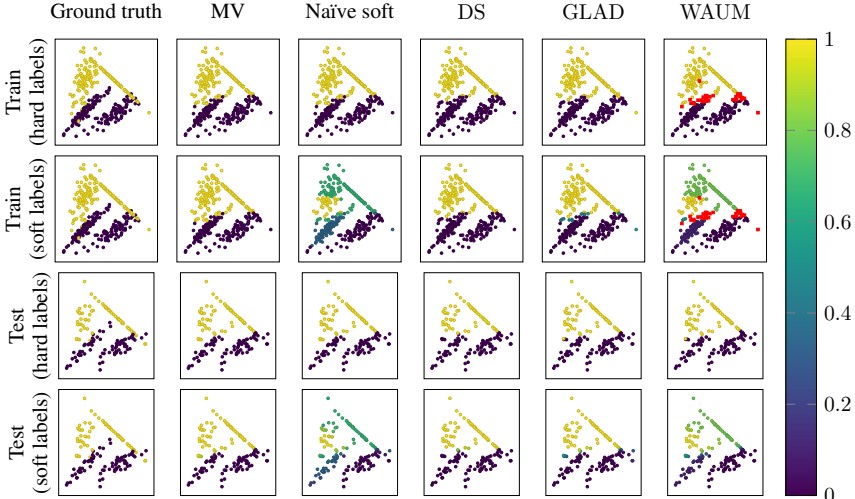

Figure 12: Training set and test predictions probabilities by a three dense layers' artificial neural network $(30, 20, 20)$ on the `make_classification`'s simulated dataset depending on the label aggregation strategy. Points in red are pruned from the training set in the WAUM aggregation. The $\alpha$ hyperparameter is set to 0.1. Each point represents a task $x_i$, and its color is the probability to belong in class 1. This lets us visualize the ambiguity in the soft training aggregated labels, but also in the resulting predictions by the neural network.

**The `two_moons` dataset: a case where pruning is not recommended** The `two_moons` simulation framework showcases the difference between relevant ambiguity in a dataset and artificial one. This dataset is created using `make_moons` function from `scikit-learn`. We simulate $n_{\texttt{task}} = 500$ points, a noise $\varepsilon = 0.2$ and use a test split of $0.3$. As can be observed with Figure 13 and Figure 14, the difficulty on this dataset comes from the two shapes leaning into one another. However, this intrinsic difficulty is not due to noise, but is actually inherent to the data. In this case, removing the hardest tasks mean removing points at the edges of the crescents, and those are in fact important in the data's structure. From Table 7, we observe that learning on naive soft labeling leads to better performance than other aggregations. But with these workers, no aggregation produced label capturing the shape of the data.

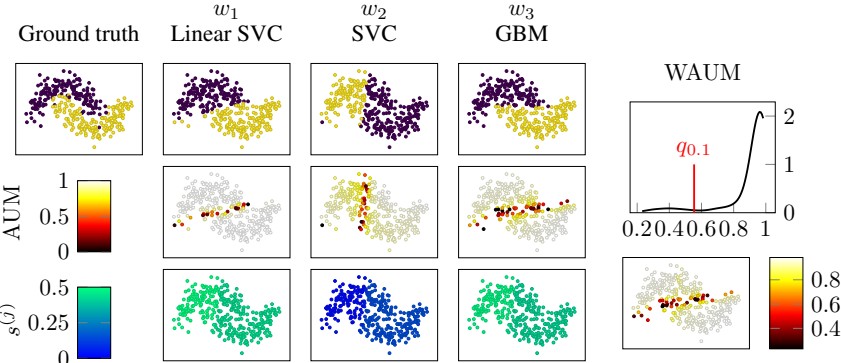

Figure 13: Simulated workers with associated AUM and normalized trust scores on the `two_moons` dataset. The hyperparameter $\alpha$ is set to $0.1$.

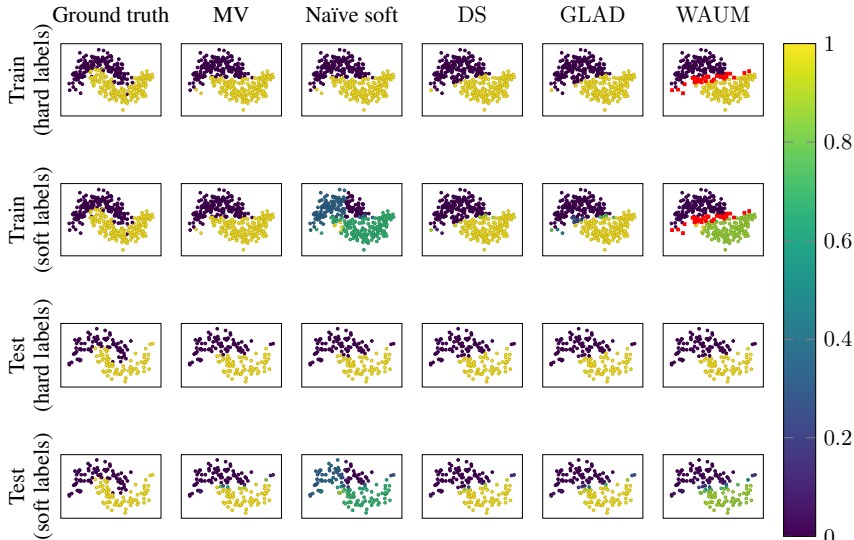

Figure 14: Training set and test predictions probabilities by a three dense layers' artificial neural network $(30, 20, 20)$ on the `two_moons` simulated dataset depending on the label aggregation strategy. Points in red are pruned from the training set in the WAUM aggregation. The $\alpha$ hyperparameter is set to $0.1$. Each point represents a task $x_i$, and its color is the probability to belong in class 1. This lets us visualize the ambiguity in the soft training aggregated labels, but also in the resulting predictions by the neural network.

Table 7: Training and test accuracy depending on the aggregation method used for the `two_moons`'s dataset with $n_{\text{task}} = 500$ points used for training a three dense layers' artificial neural network $(30, 20, 20)$. For reference, the best worker is $w_3$ with a training accuracy of $0.923$ and a test accuracy of $0.900$.

| Aggregation | $\text{Acc}_{\text{train}}$ | Test accuracy | ECE |
|---|---|---|---|
| MV | 0.917 | **0.894** $\pm$ 0.002 | **0.098** $\pm$ 0.004 |
| Naive soft | 0.917 | **0.887** $\pm$ 0.002 | 0.217 $\pm$ 0.010 |
| DS | 0.871 | 0.867 $\pm$ 0.000 | 0.126 $\pm$ 0.001 |
| GLAD | 0.006 | 0.872 $\pm$ 0.006 | 0.107 $\pm$ 0.004 |
| WAUM($\alpha = 10^{-3}$) | 0.917 | 0.875 $\pm$ 0.002 | **0.088** $\pm$ 0.012 |
| WAUM($\alpha = 10^{-2}$) | 0.919 | 0.874 $\pm$ 0.002 | **0.092** $\pm$ 0.011 |
| WAUM($\alpha = 10^{-1}$) | 0.926 | 0.870 $\pm$ 0.003 | 0.101 $\pm$ 0.020 |
| WAUM($\alpha = 0.25$) | 0.946 | 0.829 $\pm$ 0.006 | 0.135 $\pm$ 0.011 |

# E    MODELS LIMITATIONS AND POSSIBLE EXTENSIONS

## E.1    GLAD LIMITATIONS: ILLUSTRATION ON A CLASSICAL DATASET

Our goal in this section is to show the importance to consider both the tasks and the labels answered when assessing a task's difficulty in an aggregation model. Indeed, we show experimentally that removing ambiguous tasks from training leads to significant accuracy improvements for GLAD. In this experiment, we consider the classical crowdsourcing setting where no task is available (*i.e.,* no $\mathcal{X}_{\text{train}}$). We investigate the performance of the GLAD algorithm to retrieve training labels. Recall that GLAD takes into account both tasks' intrinsic difficulties and workers' abilities.

**The `three difficulties` dataset.**  We consider the `three difficulties` dataset, a crowdsourcing example with binary labels we have adapted from Whitehill et al. (2009). In this dataset, a crowd of $n_{\text{worker}} = 50$ workers has to label $n_{\text{task}} = 1000$ tasks. Labels are binary and balanced. Tasks are either `easy`, `hard` or `random` indicating their intrinsic difficulty. Tasks difficulties are drawn with probabilities respectively $p_{\text{easy}} = 0.53$, $p_{\text{hard}} = 0.27$ and $p_{\text{random}} = 0.2$. We thus simulate $(y_i^\star, d_i^\star) \in \{0, 1\} \times \{\texttt{easy}, \texttt{hard}, \texttt{random}\}$. We model each worker ability by two levels: they are independently chosen as `good` (with probability $0.75$) or `bad` (with probability $0.25$). The interplay between task difficulty and worker ability is governed by the confusion matrices presented in Table 8. In this context, the `random` tasks represent ambiguous tasks that workers labels by flipping a fair coin.

Table 8: Confusion matrices simulated with three levels of difficulty. Tasks `easy` induce no error, `random` ones are answered as a fair coin while `hard` tasks induce more confusion to `bad` workers than to `good` ones.

| `easy` $(53\%)$ | `hard` $(27\%)$ | | `random` $(20\%)$ |
|---|---|---|---|
| | `good` worker $(75\%)$ | `bad` worker $(25\%)$ | |
| $\begin{pmatrix} 1 & 0 \\ 0 & 1 \end{pmatrix}$ | $\begin{pmatrix} 0.75 & 0.25 \\ 0.25 & 0.75 \end{pmatrix}$ | $\begin{pmatrix} 0.55 & 0.45 \\ 0.45 & 0.55 \end{pmatrix}$ | $\begin{pmatrix} 0.5 & 0.5 \\ 0.5 & 0.5 \end{pmatrix}$ |

We chose a varying number of answers per task to reflect that in non-controlled experiments, the number of answers per task can fluctuate. Hence, each tasks' number of answers is drawn uniformly at random as $|\mathcal{A}(x)| \sim \mathcal{U}(\{1, \ldots, 10\})$, and then $|\mathcal{A}(x)|$ workers are chosen uniformly at random among the $n_{\text{worker}}$. We repeat the experiment 40 times and monitor the training label accuracy obtained by GLAD.

Without pruning the `random` tasks, the training accuracies lays inside $[0.838, 0.885]$ for $95\%$ of the repetitions. When pruning the random tasks, the accuracy improves and lays inside $[0.941, 0.965]$ for $95\%$ of the repetitions. Thus, identifying such tasks is important to improve training since the accuracy is sensitive to ambiguous task (such as `random` tasks).

As displayed in Figure 15a, with GLAD, the difficulty estimated $\hat{\beta}$ only discriminates `easy` tasks from the others. Besides, when a task $x$ has only received a single label ($|\mathcal{A}(x)| = 1$, see Figure 15b) the task's difficulty cannot be estimated. Thus all these tasks have nearly the same estimated difficulty. Identifying such tasks is thus essential to distinguish between `hard` (necessary in the learning

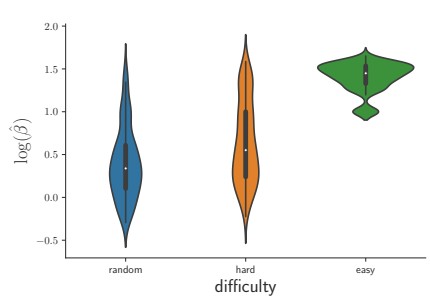 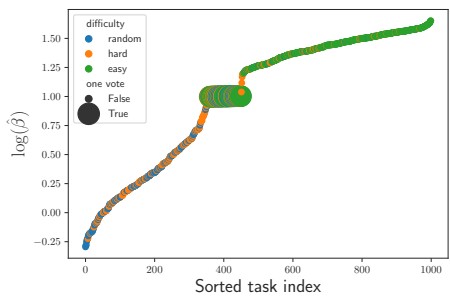

(a) Distribution of the log estimated difficulty ($\hat{\beta}$) depending in each simulated difficulty level: `random` and `hard` tasks have similar distributions

(b) Sorted tasks by estimated difficulty ($\hat{\beta}$) using GLAD: tasks with a single vote all have nearly the same estimated difficulty

Figure 15: The `three difficulties` dataset ($n_{\texttt{task}} = 1000$, $n_{\texttt{worker}} = 50$). Difficulty estimated with GLAD. Associated difficulty levels, workers quality and confusion matrices are presented in Table 8.

process) and `random` ones (that could be harmful). We have introduced the WAUM to identify such ambiguous tasks. Note that the neural network output probabilities can act as a control agent and take into account tasks that only have one vote too. This is why using models with the actual tasks $(x_i)_{i \in [n_{\texttt{task}}]}$ and not only the tasks' labels is crucial in crowdsourcing settings with high variations in the number of votes per task.

### E.2 Confusion matrices estimation in the WAUM

To compute the WAUM, we use the confusion matrices $\{\pi^{(j)}\}_{j \in [n_{\texttt{worker}}]}$ (estimated with Algorithm 1). The vanilla DS model (Dawid & Skene, 1979) can be used to estimate these confusion matrices $\pi^{(j)} \in \mathbb{R}^{K \times K}$ for each worker $w_j$. The quadratic number of parameters to estimate for each worker can lead to convergence issues with the vanilla DS model. But as stated in Section 3, any model that estimates confusion matrices can be used in the WAUM's computation.

We detail below some possible variants:

- Sinha et al. (2018), who have accelerated vanilla DS convergence by constraining the estimated labels' distribution $T_{i\bullet}$ to be a Dirac mass. Hence, predicted labels are hard labels, leading to worse calibration error than vanilla DS while preserving the same accuracy.

- Passonneau & Carpenter (2014) who have introduced Dirichlet priors on the confusion matrices' rows and the prevalence $\rho$.

- Servajean et al. (2017) who have exploits the sparsity of the confusion matrices when the number of classes $K$ is high.

- Imamura et al. (2018) estimates $L \ll n_{\texttt{worker}}$ clusters of workers thus constraining at most $L$ different confusion matrices using variational inference. This lead to estimate $K^2 \times L$ coefficients for the confusion matrices instead of $K^2 \times n_{\texttt{worker}}$.

These alternatives to the vanilla DS model could help computing faster the confusion matrices used in the WAUM for the trust score computation.

**Iterate confusion matrix estimation.** As Algorithm 1 is an iterative procedure to estimate confusion matrices $\hat{\pi}^{(j)}$ on less ambiguous tasks, one natural question is whether one iteration is enough or if repeating this process improves the performance. We denote WAUMiter the strategy that estimates the WAUM with confusion matrices obtained from the first dataset $\mathcal{D}_{\text{pruned}}$.

We consider the same workers on the `three_circles` classification task, but vary the number of votes per task. Each tasks' number of answers is drawn uniformly at random such that $1 \leq |\mathcal{A}(x)| \leq 3$. Then, $|\mathcal{A}(x)|$ workers are chosen among the $n_{\texttt{worker}}$ ones available in the crowd with probability $0.2$ for the linear SVC (that worst simulated workers), and $0.4$ for the two other classifiers. Workers'

Table 9: Performance metrics by aggregation method for the `three_circles` dataset with 250 tasks per label and $1 \leq |\mathcal{A}(x)| \leq n_{\texttt{worker}} = 3$ by a three dense layers' artificial neural network $(30, 20, 20)$. Test size is 0.3.

| Aggregation | Naive soft | DS | GLAD | WAUM | WAUMiter |
|---|---|---|---|---|---|
| $\text{Acc}_{\text{train}}$ | 0.729 | 0.659 | 0.583 | 0.667 | 0.753 |
| Test accuracy | $0.728 \pm 0.018$ | $0.644 \pm 0.027$ | $0.581 \pm 0.003$ | $0.725 \pm 0.056$ | $\mathbf{0.753} \pm 0.039$ |
| ECE | $0.146 \pm 0.023$ | $0.242 \pm 0.011$ | $0.196 \pm 0.004$ | $\mathbf{0.137} \pm 0.034$ | $0.156 \pm 0.023$ |

decisions are unchanged from Figure 5. Table 9 reports the train and test accuracy for the aggregation methods considered. DS and GLAD underperform WAUM, whose decision boundary is affected nonetheless. Note that, contrary to GLAD, the tasks' difficulty considered by the WAUM can be evaluated thanks to the tasks' features even with a single answer.

In short, taking into accounts tasks' features lets us break through one of GLAD limitations – estimating the difficulty for tasks with a single vote. We recall possible alternatives to compute faster the confusion matrix used in the WAUM statistic. Finally, iterating the confusion matrix and scores estimation in the WAUM procedure can improve performance in practice.

### E.3 STACKING WORKERS ANSWERS IN THE WAUM

As stated in Section 3, the WAUM computation in Algorithm 1 can be costly in scenarios with many workers, as it requires one neural network per worker to compute each AUM independently. To limit this issue, we propose a stacked version. It consists in computing the WAUM with $\mathcal{D}_{\text{stack}} := \{(x_i, y_i^{(j)})\}_{i \in [n_{\text{task}}], j \in \mathcal{A}(x_i)}$, the dataset obtained by stacking all workers' tasks and labels together. We denote this alternate version WAUMstack. The difference between Algorithm 1 and Algorithm 5 resides in the neural network training happening once on $\mathcal{D}_{\text{stack}}$ instead of one networks trained per $\mathcal{D}_{\text{train}}^{(j)}$. The AUM and trust scores are still evaluated worker-wise and task-wise. Moreover, using $\mathcal{D}_{\text{stack}}$ can break independence between workers. Indeed, for a given task, the trust score associated to worker $j$ depends on the answers given by all other workers. It might not be desirable: when many spammers are present in the crowd, the network's softmax output (on which the scores rely) could be corrupted.

---

**Algorithm 5:** WAUMstack computation.

**Data:** $\mathcal{D}_{\text{train}}$: tasks and crowdsourced labels, $\alpha \in [0, 1]$: cut-off proportion
**Result:** $\{\hat{\pi}^{(j)}\}_{j \in [n_{\text{worker}}]}$: estimated confusion matrices; $(\hat{y}_i)$: tasks' aggregated label
1 **Initialization:** Get confusion matrix estimation $\{\hat{\pi}^{(j)}\}_{j \in [n_{\text{worker}}]}$
2 **Train** a neural network on the stacked dataset $\mathcal{D}_{\text{stack}} = \{(x_i, y_i^{(j)})\}_{i,j}$ for $T$ epochs
3 **for** $j \in [n_{worker}]$ **do**
4 $\quad$ Compute $\text{AUM}(x_i, y_i^{(j)}; \mathcal{D}_{\text{train}}^{(j)})$ using Equation (2) for $i \in \mathcal{T}(w_j)$
5 $\quad$ Compute **trust scores** $s^{(j)}(x_i)$ using Equation (4) for $i \in \mathcal{T}(w_j)$
6 **for** *each task* $x \in \mathcal{X}_{train}$ **do**
7 $\quad$ Compute $\text{WAUM}(x)$ using Equation (3)
8 Get $q_\alpha$ the WAUM's **quantile threshold** of order $\alpha$ of $(\text{WAUM}(x_i))_{i \in [n_{\text{task}}]}$
9 Define $\mathcal{D}_{\text{pruned}} = \left\{ \left( x_i, (y_i^{(j)})_{j \in \mathcal{A}(x_i)} \right) : \text{WAUM}(x_i) \geq q_\alpha \text{ for } i \in [n_{\text{task}}] \right\}$
10 Compute $\{\hat{\pi}^{(j)}\}_{j \in [n_{\text{worker}}]}$ on tasks in $\mathcal{D}_{\text{pruned}}$
11 **Soft labels:** $\hat{y}_i = \frac{\tilde{y}_i}{\sum_{k \in [K]} (\tilde{y}_i)_k} \in \Delta_{K-1}$ with $\tilde{y}_i = \left( \sum_{j \in \mathcal{A}(x_i)} \hat{\pi}_{k,k}^{(j)} \mathbb{1}_{\{y_i^{(j)} = k\}} \right)_{k \in [K]}$ for all $x_i \in \mathcal{X}_{\text{pruned}}$

---

We simulate $n_w = 150$ workers who answer tasks from a dataset with $K = 4$ classes simulated using `scikit-learn`'s function `make_classification`; we call this dataset `make_classification_multiclass`. All simulated tasks are labeled by up to five workers among Linear SVCs, SVCs or Gradient Boosted Classifiers (GBM) chosen uniformly. In order to simulate multiple workers with some dissimilarities, we randomly assign hyperparameters for each classifier as follows:

- Linear SVC:
    - the margin `C` is chosen in a linear grid of 20 points from $10^{-3}$ to 3,
    - the maximum number of iterations ranges between 1 and 100,
    - and the `loss` function is in {`hinge`, `squared_hinge`}.
- SVC:
    - the `kernel` is in {`poly`, `rbf`, `sigmoid`} (polynomial is with degree 3),
    - and the maximum number of iterations is between 1 and 100.
- GBM:
    - the `learning_rate` is in {0.01, 0.1, 0.5},
    - the number of estimators is in {1, 2, 5, 10, 15, 20, 30, 50, 100},
    - and the maximum number of iterations is between 1 and 100.

All simulated workers are also initialized using different seeds. All hyperparameters are drawn uniformly at random from their respective set of possible values.

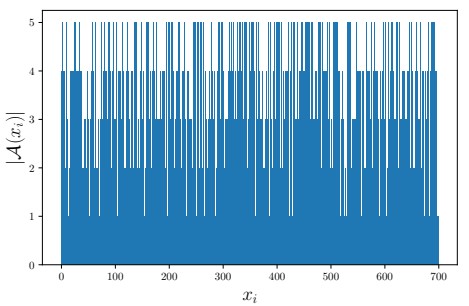
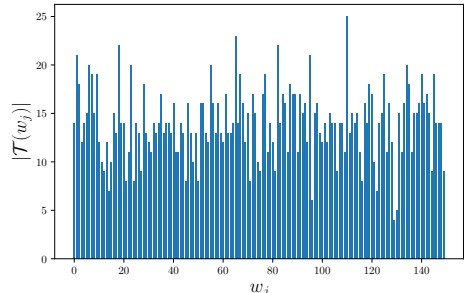

(a) Number of answers per task       (b) Number of votes per worker

Figure 16: The `make_classification_multiclass` dataset. Distribution of votes per task (a) and the number of votes for the 150 workers (b) in the simulated classification problem with $K = 4$ classes and 250 tasks per class.

Table 10 shows a case with many workers and varying number of votes ($n_{\text{worker}} = 150$ and up to five votes per task, see Figure 16 for the repartition). In this setting, the stacked version of the WAUM has the same performance as the vanilla WAUM, with a **much lower computational cost** (as we do not train $n_{\text{worker}}$ networks but a single one).

Table 10: The `make_classification_multiclass dataset`: Performance metrics by aggregation method. The number of tasks is $n_{\text{task}} = 250$ tasks per classes and $1 \leq |\mathcal{A}(x)| \leq 5$.

| Aggregation | Naive soft | DS | GLAD | WAUM | WAUMstack |
|---|---|---|---|---|---|
| $\text{Acc}_{\text{train}}$ | 0.8428 | 0.820 | 0.850 | 0.858 | 0.883 |
| Test accuracy | $0.851 \pm 0.00$ | $0.849 \pm 0.004$ | $0.842 \pm 0.002$ | $0.849 \pm 0.006$ | $\mathbf{0.861} \pm 0.007$ |
| ECE | $0.146 \pm 0.023$ | $0.242 \pm 0.011$ | $0.196 \pm 0.004$ | $\mathbf{0.137} \pm 0.034$ | $0.156 \pm 0.023$ |

## F    CIFAR-10H DATASET

Introduced by Peterson et al. (2019), the crowdsourced dataset CIFAR-10H attempts to recapture the human labeling noise present when creating the dataset. CIFAR-10H train set consists of the test set of CIFAR-10. The generalization performance is measured on CIFAR-10's training set:

$$|\mathcal{D}_{\text{train}}| = 10000 \quad \text{and} \quad |\mathcal{D}_{\text{test}}| = 50000 \ .$$

The crowdsourcing experimentation involved $n_{\text{worker}} = 2571$ workers on Amazon Mechanical Turk. Workers had to chose one of the ten labels of CIFAR-10 for each presented image (no alternative choice available). Each worker labeled 200 tasks (and was paid \$1.50 for that): 20 for each

original category. Answering time was also measured for each worker. Note that attention check occurred every 20 trials on tasks considered obvious: 14 workers failed a 75% accuracy threshold. The CIFAR-10 dataset is balanced with respect to crowd votes as each task has been labelled by 50 workers in average. However, due to the original curation of the dataset (Krizhevsky & Hinton, 2009; Aitchison, 2021) this high number of votes does not imply wide disagreements in the proposed labels. Figure 17 shows the distribution of the entropy on the soft label distribution of votes per task:

$$\forall x \in \mathcal{X}_{\text{train}}, \ \text{Ent}(x) = -\sum_{k \in [K]} \hat{y}_k \log(\hat{y}_k) \quad \text{with} \quad \hat{y} = \frac{\sum\limits_{j \in \mathcal{A}(x)} \mathbb{1}_{\{y^{(j)}=k\}}}{\sum\limits_{k \in [K]} \left( \sum\limits_{j \in \mathcal{A}(x)} \mathbb{1}_{\{y^{(j)}=k\}} \right)} \ .$$

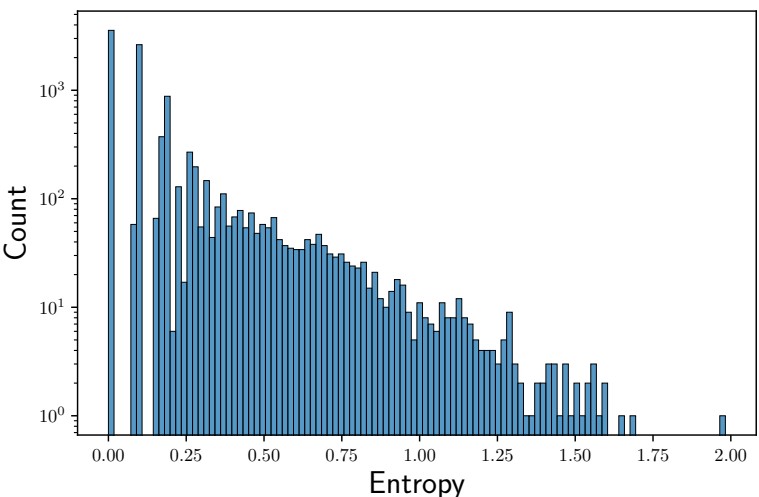

Figure 17: Frequency of task's naive soft labels entropies (in logarithmic scale) in the CIFAR-10H dataset. Only very few tasks show workers disagreeing. For reference, the maximum entropy is $\log(K) \simeq 2.303$.

## G    IDENTIFICATION AND AGGREGATION USING THE WAUM

As stated in Section 3.2, the WAUM is primarily a statistic used to identify possibly too ambiguous tasks (or even mislabelled ones) thanks to crowdsourced labels. In Algorithm 1, we present a pipeline from identification to learnable labels using the WAUM. After pruning ambiguous tasks, we chose as aggregation scheme a simple weighted distribution of votes with weights based on expertise obtained by the DS model. Yet, it is also possible to run any aggregation strategy after pruning the tasks flagged "ambiguous".

We show in Table 11 that by considering the WAUM's pruning with DS (resp. GLAD) soft labels, performance improves upon simply using DS alone (resp. GLAD alone). This empirically shows that identifying and removing these tasks can improve the learning efficiency, in agreement with already existing experiments on data cleaning and pruning by Pleiss et al. (2020) for standard supervised settings. For example, on the simulated `three_circles` dataset, the vanilla DS reached $0.753$ in test accuracy, but using pruning with the WAUM lead to a test accuracy score of $0.804$ (competitive with our first proposed aggregation method). A similar improvement can be observed with the GLAD's strategy.

We also compare our results with `Crowdlayer` (MW), a feature-aware strategy proposed by Rodrigues & Pereira (2018) to learn from crowdsourced labels using an additional layer on the network. This additional layer is used to model each worker's confusion with one confusion matrix per worker (as in the DS model). Note that this strategy does not aggregate labels, thus we can not compute the $\text{Acc}_{\text{train}}$ metric. Note other strategies (Ma & Olshevsky, 2020; Ibrahim et al., 2019; Ma et al., 2020) are not considered in this paper, due to missing open-source implementations.

Table 11: Training and test accuracy depending on the aggregation method used for the `three_circles`'s dataset with 250 tasks per class used for training a three dense layers' artificial neural network $(30, 20, 20)$. The hyperparameter $\alpha$ is set to $0.1$ for the WAUM computations.

| Aggregation | $\text{Acc}_{\text{train}}$ | Test accuracy | ECE |
|---|---|---|---|
| DS | 0.747 | 0.753 | 0.220 |
| WAUM + DS | 0.887 | **0.804** | 0.171 |
| GLAD | 0.549 | 0.578 | 0.356 |
| WAUM + GLAD | 0.569 | 0.605 | 0.333 |
| Crowdlayer (MW) | – | 0.640 | 0.234 |
| WAUM | 0.928 | **0.806** | **0.078** |

# H  USING THE LABELME DATASET

Another real dataset in the crowdsourced image classification field that can be used is the `LabelMe` crowdsourced dataset created by Rodrigues & Pereira (2018). This dataset consists of $n_{task} = 1000$ training images dispatched among $K = 8$ classes. The validation set has 500 images and the test set 1188 images. The whole training tasks have been labelled by $n_{worker} = 59$ workers, each task having between one and three given (crowdsourced) labels. In particular, 42 tasks have been labelled only once, 369 tasks have been labelled twice and 589 received three labels. This is a way sparser labelling setting than the CIFAR-10H dataset. Following Appendix E.1, GLAD's algorithm fails at inferring a difficulty parameter on such tasks.

In our experiment, we train a VGG-16 (Simonyan & Zisserman, 2014) with batch normalization on the `LabelMe` dataset for 100 epochs using the same label aggregation strategies as in Appendix A, with the same hyperparameters as in Section 4. The neural network is initialized with weights from training using `ImageNet` dataset. The learning rate is multiplied by a factor 0.1 at epochs 50 and 75. The WAUM is thus computed on the first 50 epochs.

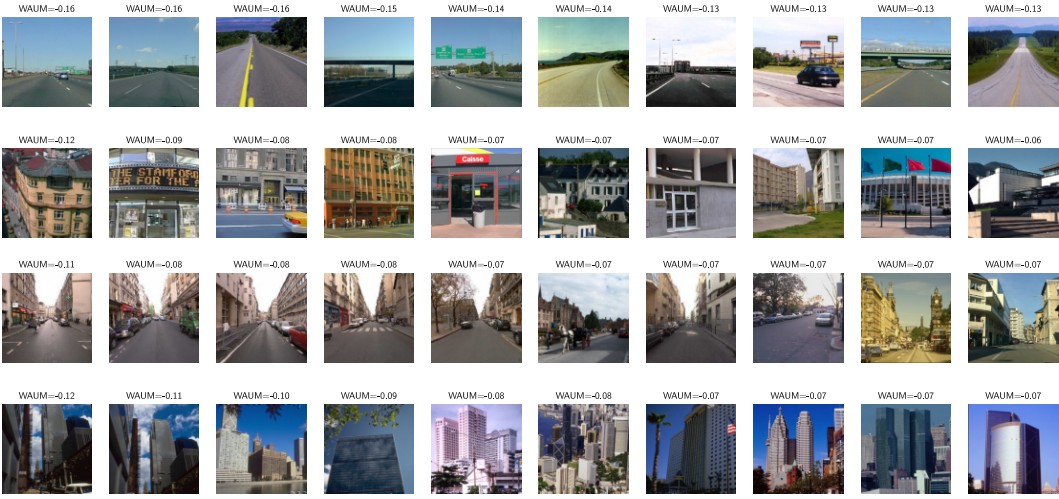

Figure 18: `LabelMe`: top-10 worst images detected in `street`, `insidecity tallbuilding` and `highway` classes using the WAUM. Overlapping classes lead to labelling confusion and learning difficulties for both the workers and the neural network.

The `LabelMe` dataset has classes that overlap and thus lead to intrinsic ambiguities. For example, the classes `highway`, `insidecity street` and `tallbuilding` (in rows) are overlapping for some tasks. Most cities have streets and tall buildings, and highways are a specific type of street, so the confusion naturally happens as can be seen in Figure 18. After training, from Table 12 we observe that the proposed feature aware aggregation using the WAUM lead to better performance in test accuracy and calibration.

Table 12: Training and test accuracy depending on the aggregation method used for the `LabelMe`'s dataset with $n_{task} = 1000$, training a VGG-16 (here with $\alpha = 0.01$ for the WAUM). Results are averaged over 5 repetitions.

| Aggregation | $Acc_{train}$ | Test accuracy | ECE |
|---|---|---|---|
| MV | 0.764 | $0.662 \pm 1.018$ | $0.143 \pm 0.005$ |
| Naive soft | 0.764 | $0.723 \pm 3.074$ | $0.138 \pm 0.013$ |
| DS | 0.797 | $0.705 \pm 3.637$ | $0.141 \pm 0.015$ |
| GLAD | 0.776 | $0.736 \pm 3.868$ | $0.134 \pm 0.004$ |
| Crowdlayer (MW) | – | $0.733 \pm 0.078$ | $0.166 \pm 0.058$ |
| WAUM | 0.775 | $\mathbf{0.742} \pm 2.917$ | $\mathbf{0.128} \pm 0.009$ |

