# OpenReview forum: "Improve learning combining crowdsourced labels by weighting Areas Under the Margin"
_ICLR.cc/2023/Conference — Submitted to ICLR 2023_

### Official Review · Reviewer_JrVm · 2022-10-23

**Confidence:** 5
**Correctness:** 3
**Technical Novelty And Significance:** 2
**Empirical Novelty And Significance:** 2
**Recommendation:** 5

**Clarity, Quality, Novelty And Reproducibility:**

This study is intuitive and technically sound. In addition, the paper is presented clearly and easy to follow and reproduce. The method is simple but interesting and seems effective, but the novelty is limited.

**Strength And Weaknesses:**

This study is intuitive and technically sound. The proposed method is simple but interesting. In addition, the paper is presented clearly and is easy to follow and reproduce.

However, there are several weaknesses existing in this paper:

- First, the main idea of this paper is straightforward and not very impressive, and the novelty of the method is limited. The proposed WAUM seems like a simple combination of existing works, i.e. AUM and confusion matrix (which is mentioned in many EM-based label aggregation methods). I suggest the authors discuss the contribution of this paper in more detail and highlight the innovations.
- Second, the design of the experiments is somewhat weak. 1) The proposed WAUM-based method measures the task difficulty by incorporating feature information and trains a neural network classifier simultaneously, but the comparison methods such as MV, DS, and GLAD are all feature-blind methods, also the comparison methods are classical algorithms but not the latest research works, so the experimental results are less convincing. I suggest that more algorithms (better to use the feature of tasks) should be compared with the proposed methods, e.g., (Raykar et al, 2010), (Rodrigues et al, 2018), and so on.
- Furthermore, since the main idea of the proposed method is to identify the harmful tasks and prune them to reconstruct a new sub-dataset, could the sub-dataset be used on other existing methods to show the effectiveness of the pruned dataset?

Raykar VC, Yu S, Zhao LH, Valadez GH, Florin C, Bogoni L, Moy L (2010), Learning from crowds. The Journal of Machine Learning Research 11:1297–1322.
Filipe Rodrigues and Francisco C. Pereira. Deep learning from crowds. AAAI’18, 2018.


**Summary Of The Paper:**

This paper proposes a novel supervised learning model in the scenarios of crowdsourcing based on AUM, which is a confidence indicator for each task. The proposed method generalizes the existing AUM indicator to the crowdsourcing setting as WAUM (the Weighted Area Under the Margin), and the WAUM identifies harmful data and prunes the ambiguous tasks to improve the results of label aggregation. Experimental results show the effectiveness of the proposed method on several simulated and real-world datasets.

**Summary Of The Review:**

Overall, the paper has its merits to show a new method introducing a crowdsourcing-specific confidence indicator to prune harmful tasks and improve the performance of the model trained on the pruned dataset. However, it also has  some  weaknesses. Thus, this paper cannot meet the high-quality requirements of ICLR.

---

> ### Author Response · Authors · 2022-11-14
> **Response to reviewer JrVm**
>
> Thank you for your comments, here is our response.
>
> 1. Simplicity:
> The simple design and intuitive comprehension is one of our main goal. Crowdsourcing methods evaluate workers answers and create trust scores. We dread that complicated or non-interpretable trust scores would prevent practitioners from adopting them. In our case, we based the WAUM on the DS model (popular, effective and easy to interpret), and on the AUM by Pleiss et al. (2020) which is also an intuitive statistic for quantifying learning difficulty.
>
> 2. Other methods:
> The standard DS model relies on estimating a confusion matrix.
> The WAUM can potentially fit with any strategy that is also based on such confusion matrix (like CrowdLayer by Rodrigues & Peirera (2018) for example).
> Once ambiguous tasks have been identified by the WAUM (what can be seen as a data cleaning step), we then chose to prune them and re-run a DS (confusion matrix) estimation procedure.
> Yet, one could run any other similar method on such a cleaned dataset.
> We have added an experiment to show learning improvements by including such a pruning step before using either GLAD or DS (see Table 11 appendix G).

---

> > ### Comment · Reviewer_JrVm · 2022-11-20
> > **Rebuttal Read**
> >
> > Thank authors for providing feedback on my concerns. However, considering the weak novelty of this paper, I would not like to change my rating.

---

> > > ### Author Response · Authors · 2022-11-24
> > > **Re: Rebuttal Read**
> > >
> > > Thank you for your reply and acknowledging that the modifications you required were incorporated in the current version of the paper.
> > > Concerning the novelty of our contribution, we would like to point out that two other reviewers wrote "The proposed method is [...] new as far as I can tell." and "Novelty seems to be high".
> > >
> > > This work is an extension of the AUM by Pleiss et. al (2020) to the crowdsourcing setting. In addition to the modifications proposed in the current version, it is currently unclear what fails to meet your expectations. Could you clarify that point?

---

### Official Review · Reviewer_jfcJ · 2022-10-25

**Confidence:** 5
**Clarity, Quality, Novelty And Reproducibility:** Please see the detailed comments above.
**Correctness:** 4
**Technical Novelty And Significance:** 3
**Empirical Novelty And Significance:** 3
**Recommendation:** 6

**Strength And Weaknesses:**

This paper is well-organized, the structure is clean and clear, and some nice graphs are provided to help the readers to understand the proposed framework. Also, the authors provided links for almost all the involved datasets and baseline methods, which brings convenience to the readers and shows respect to the original authors at the same time. However, this work can be further improved in the following aspects:

- The baseline methods used are classic methods but not very new. It would be better to use some more recent methods as baseline methods.
- Since the proposed algorithm used the idea of AUM method, can you compare these two methods in depth?
- In this paper, the comparison with some vote aggregation algorithms is provided, however, these methods are not new, can you compare the proposed method with more recent aggregation methods [1][2][3]
[1] https://jmlr.org/beta/papers/v21/19-359.html
[2] https://arxiv.org/pdf/1909.12325.pdf
[3]https://proceedings.neurips.cc/paper/2020/file/f86890095c957e9b949d11d15f0d0cd5-Paper.pdf
- The format of the tables used in this paper should meet the requirement of the conference.
- It would be better to give some theoretical insights or analyses why the proposed algorithm works well.


**Summary Of The Paper:**

This work considers the crowdsourcing problems where there exist ambiguous tasks. To address this problem, a new framework that used a similar technique in AUM method to identify ambiguous tasks has been proposed. To show the effectiveness of the proposed method, evaluations on both the synthetic datasets and the CIFAR-10H crowdsourced dataset have been implemented.

**Summary Of The Review:**

In my opinion, this paper is a borderline paper. The motivation is valid, and the organization is clear. However, the baseline methods used are not very new which is my concern.

---

> ### Author Response · Authors · 2022-11-14
> **Response to Reviewer jfcJ**
>
> Thank you for your valuable comments. We have taken them into account as follows:
>
> 1. New methods:
> The chosen baseline are voluntarily standard.
> Indeed, our method is flexible and can be combined with any method that estimates a score per worker (or that outputs an estimation of the labels confusion matrix, as we do) can leverage our framework. We have hence used standard methods in the literature to illustrate that. In particular, we have added an experiment showing that pruning with the WAUM can improve both GLAD and DS aggregations.
> We also want to emphasize that most recent methods are based either on DS or GLAD's models that we compare with.
> The Crowdlayer method by Rodrigues & Pereira (2018) can be combined with the WAUM, as it outputs a confusion matrix. The code being public, we have conducted some comparisons with its MW version, results on our simulated three_circles dataset are provided in Table 11, Appendix G (and are a bit behind the performance of DS). Results on LabelMe should be updated by the end of the week, results on CIFAR-10H might take longer and might not be available by the end of the rebuttal period.
> Thanks for pointing out references [1,2,3]. We have added them to our references and cited them appropriately. Yet, they are methods that do not use the tasks/images themselves but only the labels.
> They aim at estimating an improved score for each worker's skill. To the best of our knowledge, they also do not provide Python code we could compare with, hence we have discarded them. Let us know if you are aware of implementations we could be re-use.
>
> 2. Difference with the original AUM:
> As answered to Reviewer Zduv, we have added a comparison with the original AUM margin.
> Note that we did not emphasize the difference (they are tiny), but we want here to motivate our proposition: the Yang & Koyejo's approach is better suited for generalization to top-K classification.
> In terms of interpretation, the margin used in our context does not take into account workers who are as confused as the neural network between the first 2 classes.
>
> 3. Format issue:
> Can you clarify the issue on the table format? We believe our tables satisfy the ICLR rules, but we might have missed one.
> 4. Theory is missing:
> We agree our work is not theoretically grounded. Our goal is to adapt the AUM in the crowdsourced setting. Even a theoretical analysis of the original AUM is still missing and appears to be out of the scope of this paper.

---

### Official Review · Reviewer_Zduv · 2022-10-27

**Confidence:** 3
**Correctness:** 2
**Technical Novelty And Significance:** 3
**Empirical Novelty And Significance:** 4
**Recommendation:** 5

**Clarity, Quality, Novelty And Reproducibility:**

Clarity is high. The paper is well organized, but I some important contents are included in the supplementary sections, such as Section B. I did not understand the motivation of the paper (see the previous section on "Strength And Weaknesses").

Quality is high. The experiments are based on a single dataset that does not have many ambiguous tasks which does not seem to be a good fit (however, I also understand that there are not many datasets with multiple labels per tasks, which is also discussed in the final section of the paper.) The analysis and visualization with simulated datasets is helpful to gain a better understanding of the proposed method.

Novelty seems to be high. It is based on a very recent paper that was published in Dec. 2020 and I have not heard of a weighted version of the AUM with crowdsourced labels.

The optional reproducibility statement was not provided in the paper. There are some details of the experiments written throughout the paper. The code was provided as a supplementary file, but I did not read the code.

**Details Of Ethics Concerns:**

Although optional, I recommend adding a ethical statement section at the end.

**Strength And Weaknesses:**


Strengths

1. Taking into account task ambiguity is an important research topic.
2. The proposed method can achieve a higher test accuracy for the simulated synthetic datasets and achieve the same performance as the naive soft method in the CIFAR-10H dataset.

Questions (and Weaknesses)

1. One of the motivation of the paper is to remove intrinsically ambiguous tasks (which might even fool expert workers) because these tasks may be harmful for the learning step. My understanding is that intrinsically ambiguous tasks are helpful for learning, and I was not sure why these tasks may be harmful. For example, the most intrinsically ambiguous task may have p(y=+1|x) = p(y=-1) = 0.5 in binary classification (assuming y \in {+1, -1}), where p(y|x) is the underlying true class probability. If we collect many hard labels (n labels) for this same task, close to half of them will be positive hard labels and the rest of the half will be negative hard labels. When we naively regard this as n samples of (x,y) pairs (as well as other tasks), and train a classifier, our (probabilistic) classifier will try to learn hatp(y|x) = 0.5 for this task, and perhaps this will lead to improved ECE. It will do no harm for generalization performance, because test samples close to this x is intrinsically hard. On the other hand, I understand that noisy mislabelled samples (which is not related to the intrinsic difficulty of the task) may be harmful for learning. In Section 3 of the AUM paper by Pleiss et al. (2020), the following sentence "In general, we assume both easy and hard correctly-labeled samples in D_train improve model generalization, whereas mislabeled examples hurt generalization" is similar to my understanding. It would be helpful if the paper can provide some discussions about why removing intrinsically ambiguous tasks will be beneficial.
2. The paper explains that the margin in Eq. 2 is modified from the original version, following Yang & Koyejo (2020). I did not have time to fully go through Yang & Koyejo (2020), but I am predicting that this corresponds to using softmax_y - softmax_2 in Eq. 2 instead of softmax_y - max_{j\neq y} softmax_j which is used in Pleiss et al. (2020), assuming we ignoring the softmax-logit difference. I wasn't sure if the nice theoretical properties discussed in Yang & Koyejo (2020) directly transfer to AUM as a mislabel identification metric. Intuitively, the original margin definition seems to be easier to understand, e.g., if we consider the right example in Fig.2 of Pleiss et al. (2020), the modified margin will be near zero because Dog logit is the 2nd largest value for most of the epochs. It would be helpful if the paper can further discuss the benefits of the modified margin.
3. Is it also possible to compare the unmodified and modified margins in the experiments?
4. The paper explains that CIFAR-10 has very few (intrinsically) ambiguous tasks, because of the careful creation of the dataset. Since the goal of the proposed method is to remove ambiguous tasks, is this a good benchmark dataset to use to test the perfomance of the proposed method?

Minor comments, suggestions
1. Page 4, typo: "Appendix A" should be "Appendix B".
2. What is bold in Table 2, 3, and 4? Are they results of a t-test?
3. Is it possible to provide a practical suggestion for the number of hard labels per task?

**Summary Of The Paper:**

This paper proposes a way to combine crowdsourced multiple labels per task by using a weighting areas under the margin method (WAUM), which uses area under the margin method (AUM) used for detecting mislabeled data in the training dataset with a single hard label per task. It is a weighted version of AUM based on the trust factor. The trust factor can be derived as the inner product between the diagonal of the confusion matrix and the softmax vector. Experiments show how the proposed method is helpful in achieving higher test accuracy and lower ECE scores.

**Summary Of The Review:**

The paper works on an interesting problem and provides a novel algorithm that aims to achieve better generalization performance. However I had a hard time understanding the main motivation of the paper along with some other questions/weaknesses. More discussions around those questions would be helpful for the reader of the paper.

---

> ### Author Response · Authors · 2022-11-14
> **Response to Reviewer Zduv**
>
> Thanks for the comments.
> We first would like to emphasize the main point of our work is to improve hard task identification. Then, this identification could be leveraged for different applications. We have considered a case where this is used to clean the dataset for improved learning performance.
>
> 1. Removing ambiguous tasks:
> First, thanks for bringing up this point.
> Following Pleiss et al. (2020), we argue that removing some images can help the training (see for instance their Fig. 5 on CIFAR-10 and CIFAR-100, where taking alpha=1%in the classical AUM lead to almost 2pt of test error improvement).
> This happens when you drop noisy images, for instance images with wrong or non-appropriate labels, like some given in Figure S4 in Pleiss et al. (2020); see also our new Figure 8 in the Appendix B.1.
> It is true that in a binary setting with **many** collected labels the resulting distribution p(y|x) should be uniform over the labels and then the network can learn this uncertainty.
> However, in practical cases **only few** labels are collected for a given task. For example the LabelMe dataset (now added, cf. Table 12 Appendix H) has between 1 and 3 answers per task among 8 possible classes.
>
> 2. Margin modification:
> You are right about the Yang & Koyejo margin differing with Pleiss et al.'s margin.
> The main benefit of Yang & Koyejo's margin is mostly for extending to top-k classification (as stated in our conclusion). We now clarify this point upfront in the text.
> A more important feature we want to emphasize is the need to go from logit to probability estimates when combining workers AUM. Indeed, a normalization crucial to handle heterogeneity in crowdsourcing: the logit version vary on different scales for various workers (remind also that the logit scores could be shifted by a constant factor without modifying the probabilities after the softmax step).
> We have added an experiment (see Fig. 8 in Appendix B.1) to compare the original (both for AUM and WAUM), the formulation from Pleiss et al. (2020) and the one we propose, adapted from Yang & Koyejo (2020).
> We agree that the differences is light in this context, but we expect that our proposition would be better suited to top-k losses, thanks to the stronger theoretical properties proved by Yang & Koyejo (2020).
>
> 3. CIFAR-10H dataset:
> We are aware of the limitations of using CIFAR-10H as it is too clean. This is why we ran extensive simulations describing cases with different numbers of classes / number of votes  per task.
> Thanks to Reviewer jfcJ, we have become aware of the crowdsourced version of the LabelMe dataset. We have added an experiment on LabelMe, showing again the competitiveness of our method. We are happy to investigate more if you know others openly available.
>
> Minor issues:
>
> - Page 4, typo: “Appendix A” should be “Appendix B”:
> Thank you it is now corrected.
>
> - What is bold in Table 2, 3, and 4? Are they results of a t-test?
> Those are simply best and 2nd best when relevant.
>
> - Is it possible to provide a practical suggestion for the number of hard labels per task?
> Thanks for raising this interesting question on the design of experiment describing how labels are collected for each task. Yet, this difficult issue is independent of our contribution, and we do not address it directly in the paper.
> However, we believe our proposed WAUM could be relevant in a sequential crowdsourcing setting. Indeed, it could help to identify tasks requiring additional expertise, and guide how to allocate more experts/workers for such identified tasks.
> Note also that the number of tasks that need more labeling would be controlled by our alpha hyperparameter.

---

### Official Review · Reviewer_Q6Gx · 2022-10-29

**Confidence:** 4
**Correctness:** 4
**Technical Novelty And Significance:** 3
**Empirical Novelty And Significance:** 3
**Recommendation:** 6

**Clarity, Quality, Novelty And Reproducibility:**

The article is particularly well written, very clear, with references to the main articles in the literature on the subject. The appendices detail the state of the art papers used in the comparisons and give further analysis of the results.

**Strength And Weaknesses:**

* The proposed method is motivated, very clearly described and new as far as I can tell.
* As far as the performance of the proposed method is concerned, the experiments do not really allow to highlight a major contribution. As indicated by the authors, the CIFAR-10H dataset may already be too clean to show the contribution of the proposed method.

It seems to me that the results should be confirmed on other real datasets, as indicated by the authors in their conclusion to give more strength to their proposal.



**Summary Of The Paper:**

This paper proposes to combine a measure of the difficulty of an example in a supervised classification task (AUM) with the performance of annotators, calculated from their confusion matrix on a crowd-sourcing task. This combined estimator, called WAUM (weighted AUM), allows to take into account both the intrinsect difficulty of an example and the confidence in its label, when the latter comes from a crowd-sourcing campaign. The method is described in detail and compared to four other standard methods in the literature. The experiments are conducted on 2 synthetic datasets and 1 real dataset (CIFAR-10H). The WAUM method achieves slightly better classification results than the other methods on the synthetic data set and similar to the other methods on the real data set.



**Summary Of The Review:**

The proposed metric developed in the paper is well motivated and clearly described. The experimental evaluation is still limited and has not been able to show performances clearly superior to the state of the art. The results described in the article still seem too preliminary.

---

> ### Author Response · Authors · 2022-11-14
> **Response to Reviewer Q6Gx**
>
> We thank the reviewer for the encouraging evaluation and constructive comments. We address the points that were raised in the review below.
>
> 1. Other real datasets for experimental validation: As mentioned in the paper, apart from CIFAR-10H, few image classification crowdsourced datasets have released both tasks and proposed labels. We will add the crowdsourced subset of LabelMe introduced in Rodrigues & Pereira (2018) raised by Reviewer Zduv. This weakness was acknowledged from our side, hence the different simulations shown in the paper (and in appendix) with several datasets and varying number of labels.
> 2. Performance: The motivation of our introduced WAUM statistic is twofold: **a**) identify tasks that are too hard for workers to identify (e.g., those shown as red points in the simulated datasets from Fig. 4 and 5) and **b**) improve learning performance w.r.t. other aggregations schemes when using a dataset with such points removed. Even if we emphasized mostly the second part in our work, the first one could be of independent interest, in particular to identify early the images that need extra labelling efforts, or that cannot be correctly labeled at all.

---

### Author Response · Authors · 2022-11-14
**Global Response**

First and foremost, we thank all reviewers for the involvement and the points highlighted to improve this paper.

We have added extra experiments to enforce our main message on the importance of **detecting hard tasks** (or wrongly labelled in original datasets). In particular, we show visually the benefits of introducing the WAUM w.r.t. the AUM (see new Figures 8 in the Appendix B.1) for that.

We have also clarified the improvements obtained by the **pruning step** (i.e., remove spurious tasks based on WAUM scores) performed before aggregating the labels. We now also show learning improvements by including the pruning step before using either GLAD or DS, illustrating the versatility of the WAUM.

As stated in our specific answers (see below), we have **added another real crowdsourced dataset**, suggested by one of the reviewers (LabelMe), and an **additional feature aware method**, Crowdlayer, by Rodrigues & Peirera (2018). The corresponding results are synthesized in Table 11 (resp. 12) in the appendix G (resp. H), and prove the competitiveness w.r.t. Crowdlayer.

A **comparison between the original margin and the one we consider** is also provided to clarify points raised by Reviewer Zduv and jfcJ. We reckon that the difference is narrow, but do not claim that this is a particularly important contribution. Our motivation, was merely to propose a modification for future generalization to top-k losses.


We now state precise answers for each reviewer's comments.

---

### Author Response · Authors · 2022-11-17
**Global response with added dataset and method**

The updated version of the manuscript now contains a **comparison on the real crowdsourced dataset LabelMe** by Rodrigues et al (2018)  with the competitive **Crowdlayer feature aware learning strategy** by Rodrigues et al (2018).
Our method using the WAUM obtains competitive performance both in calibration and testing accuracy (Table 12 in appendix H) and identifies ambiguities within the LabelMe dataset (Figure 18 appendix H).

We are wondering if the revised version with an added real experiment and competitive feature aware method answers your concerns.
We are pleased to clarify any question that you might have and hope that you might take these new modifications into account for the scores.

Thank you for your time and we look forward to hearing back.

---

### Author Response · Authors · 2022-12-07
**acknowledging new version and discussion**

Dear Reviewers,

The end of the discussion phase is coming soon, and we have not yet heard back from most of you.
We are wondering if you have further questions or concerns regarding our response and the updated manuscript.
We hope our reply addressed your concerns and that you could reconsider your score accordingly.

Thanks again for your feedback.
The authors.

---

### Decision · Program_Chairs · 2023-01-20

**Decision:**

Reject

**Justification For Why Not Higher Score:**

A higher score could have been given if the experimental results had shown larger differences with the baselines on the real dataset, or if the authors had articulated a rationale for the approach, for instance supported by a theoretical analysis under reasonable assumptions.

**Justification For Why Not Lower Score:**

N/A

**Metareview: Summary, Strengths And Weaknesses:**


The reviewers agreed that the paper presents an interesting approach, in a paper that is well written. However, the main weaknesses of the paper -- fairly limited technical novelty (combining AUM with weighting techniques), lack of theoretical support or strong intuitive motivation for the approach, as well as limited experimental results (small improvements and limited baselines) remained despite the additional experiments provided in the rebuttal.

**Summary Of Ac-Reviewer Meeting:**

The paper was considered borderline since the experimental validation was consistently considered a weakness of the paper by the reviewers, but the authors provided additional experiments and results during the rebuttal.

The points raised in the meeting concerned the new experiment, novelty and strength of the results.

Overall, the new experiments were acknowledged, but reviewers remained unconvinced by the results and the baselines.

- on the novelty side, the ad-hoc/heuristic nature of the approach was considered a weakness of the paper by all present reviewers.

- following up on Rev jfcJ's review, more recent baselines could still be tried out at little cost, even if it meant coding one (some are fairly simple)

- the new experimental results described in Appendix H show significant but small improvements (~1% of accuracy), not really a game-changer. These results were considered not sufficient to compensate for the lack of strong motivation for the approach.